# Feedbacks between the formation of secondary minerals and the infiltration of fluids into the regolith of granitic rocks in different climatic zones (Chilean Coastal Cordillera)

Ferdinand J. Hampl[1], Ferry Schiperski[1], Christopher Schwerdhelm[2], Nicole Stroncik[3], Casey Bryce[4], Friedhelm von Blanckenburg[3,5], Thomas Neumann[1]

[1]Department of Applied Geochemistry, Technische Universität Berlin, Ernst-Reuter-Platz 1, 10587 Berlin, Germany
[2]Geomicrobiology Group, Eberhard-Karls-University Tübingen, Schnarrenbergstrasse 94-96, 72076 Tübingen, Germany
[3]Earth Surface Geochemistry, GFZ German Research Centre for Geosciences, Telegrafenberg, 14473 Potsdam, Germany
[4]School of Earth Sciences, University of Bristol, Wills Memorial Building, Queens Road, Bristol BS8 1RJ, United Kingdom
[5]Institute of Geological Sciences, Freie Universität Berlin, Malteserstrasse 74-100, 12249 Berlin, Germany

*Correspondence to*: Ferdinand J. Hampl (ferdinand.j.hampl@tu-berlin.de)

**Abstract.** Subsurface fluid pathways and the climate-dependent infiltration of fluids into the subsurface jointly control the intensity and depth of mineral weathering reactions. The products of these weathering reactions (secondary minerals), such as Fe(III) oxyhydroxides and clay minerals, in turn exert a control on the subsurface fluid flow and hence on the development of weathering profiles.

We explored the dependence of mineral transformations on climate during the weathering of granitic rocks in two 6 m deep weathering profiles in Mediterranean and humid climate zones along the Chilean Coastal Cordillera. We used geochemical and mineralogical methods such as (micro-) X-ray fluorescence, oxalate/dithionite extractions, X-ray diffraction and electron microprobe mapping to elucidate the transformations involved during weathering. In the profile of the Mediterranean climate zone, we found a low weathering intensity affecting the profile down to 6 m depth. In the profile of the humid climate zone, we found a high weathering intensity. Based on our results, we propose mechanisms that can intensify the progression of weathering to depth. The most important is weathering-induced fracturing (WIF) by Fe(II) oxidation in biotite and precipitation of Fe(III) oxyhydroxides, and by swelling of interstratified smectitic clay minerals that promotes the formation of fluid pathways. We also propose mechanisms that mitigate the development of a deep weathering zone, like the precipitation of secondary minerals (e.g., clay minerals) and amorphous phases that can impede the subsurface fluid flow. We conclude that the depth and intensity of primary mineral weathering in the profile of the Mediterranean climate zone is significantly controlled by WIF. It generates a surface-subsurface connectivity that allows fluid infiltration to great depth and hence promotes a deep weathering zone. Moreover, the water supply to the subsurface is limited in the Mediterranean climate and thus most of the weathering profile is generally characterized by a low weathering intensity. The depth and intensity of weathering processes in the profile of the humid climate zone, on the other hand, are controlled by an intense formation of

secondary minerals in the upper section of the weathering profile. This intense formation arises from pronounced dissolution
of primary minerals due to the high water infiltration (high precipitation rate) into the subsurface. The secondary minerals, in
turn, impede the infiltration of fluids to great depth and thus mitigate the intensity of primary mineral weathering at depth.
These two settings illustrate that the depth and intensity of primary mineral weathering in the upper regolith are controlled by
positive and negative feedbacks between the formation of secondary minerals and the infiltration of fluids.
**Keywords:** Coastal Cordillera, feedback, weathering-induced fracturing, secondary minerals, Critical Zone, fluid flow

## 41 1 Introduction

The formation of weathered material (regolith) from unweathered rock (bedrock) is a key process for shaping Earth´s surface.
It is of major importance for making mineral-bound nutrients accessible to the biosphere of the Critical Zone (e.g., Dawson et
al., 2020) and to supply rocks and minerals to the sediment cycle. In this process the in-situ disaggregation and chemical
depletion of weathered rock (saprock) to saprolite plays an essential role. This transformation is a result of fracturing and
mineral dissolution (e.g., Navarre-Sitchler et al., 2015). Both are associated with chemical, physical (e.g., Goodfellow et al.,
2016), and biological weathering processes (e.g., Drever, 1994; Lawrence et al., 2014; Napieralski et al., 2019). These
processes are linked to climate-related parameters such as precipitation rate, fluid flow (water and gases), and biological
activity. Apart from that, the weathering processes and hence the saprolite formation also depend on primary fractures (e.g.,
Molnar et al., 2007; Hynek et al., 2017; Kim et al., 2017; Holbrook et al., 2019; Hayes et al., 2020; Krone et al., 2021; Hampl
et al. 2022a), discontinuity density and tortuosity (Israeli et al., 2021), thermoelastic relaxation (e.g., Nadan and Engelder,
2009) as well as the topographic surface profile (e.g., Rempe and Dietrich, 2014; St. Clair et al., 2015). However, one of the
most fundamental parameters for the regolith formation is the mineral content of the bedrock. The weathering of some of these
primary minerals and the consequent formation of secondary minerals can lead to an amplification of the depth and intensity
(i.e., the parameter describing the elemental loss and relative amount of secondary minerals) of primary mineral weathering
(e.g., Fletcher et al., 2006; Lebedeva et al., 2007; Buss et al., 2008; Behrens et al., 2015; Hampl et al. 2022a). Such mechanisms
comprise (1) a forcing process like the formation of secondary minerals that is triggering (2) a responsive process such as more
intense infiltration of fluids to depth. The latter process reinforces the initial forcing process of secondary mineral formation.
Such a mechanism is therefore called positive feedback between (1) and (2). The formation of secondary minerals can also
have a weathering-impeding effect (e.g., Lohse and Dietrich, 2005; Navarre-Sitchler et al., 2015; Kim et al., 2017; Gerrits et
al., 2021) causing a mitigation of the weathering depth and -intensity. Such mechanisms comprise (1) a forcing process like
the formation of secondary minerals and (2) a responsive process such as reduced infiltration of fluids to depth. The latter
process damps the initial forcing process of secondary mineral formation, and the mechanism is therefore called negative
feedback between (1) and (2).
Deciphering the relationship between the formation of secondary minerals and the climatic conditions they were formed under
is a prerequisite for understanding the weathering system. It allows to determine whether feedbacks between the formation of
secondary minerals and the infiltration of fluids affect the intensity and depth of primary mineral weathering. We hypothesize
that a positive feedback loop results in a deep weathering depth, as secondary minerals form fluid pathways by fracturing due
to volume increase. On the other hand, we think that a negative feedback loop leads to a shallow weathering depth, as the
precipitation of secondary minerals seals fluid pathways.
To explore such connections and to elucidate the impact of secondary minerals on the development of weathering systems in
different climatic zones, we investigated two 6 m deep weathering profiles in the Chilean Coastal Cordillera. One profile is
located in a Mediterranean (mean annual temperature: 14.9 °C, mean annual precipitation: 436 mm yr$^{-1}$) and another in a humid
climate zone (mean annual temperature: 14.1 °C, mean annual precipitation: 1084 mm yr$^{-1}$) (Scheibe et al., 2023), and both
developed from weathering of granitic rock. Both sites are eroding and the surfaces in the locations are thus constantly turned
over (see compilation of rates and environmental parameters in Oeser and von Blanckenburg (2020) and references therein).
The profiles were sampled in soil pits and complemented with rock samples obtained by deep wireline rotary drilling close to
the soil pits. Samples were investigated by a combination of analytical techniques such as X-ray fluorescence and micro-X-
ray fluorescence as well as oxalate-/dithionite extraction to characterize the geochemical composition, and X-ray diffraction,
magnetic susceptibility measurements, electron microprobe as well as light microscopy to identify the mineral assemblages.
The combined results of these techniques are used to derive weathering-intensifying and -mitigating processes during
subsurface weathering and to elucidate how these processes influence the depth and intensity of weathering in the different
climate zones.
**2 Study sites**
**2.1 La Campana (LC)**
The soil pit (-33.02833° N, -71.04370° E, 894 m) and the drilling site some 15 m next to it (-33.02833° N, -71.04354° E,
898 m) are located south of the La Campana National Park approximately 60 km NW of Santiago de Chile (Fig. 1a). They are
situated on a ridge with steep slope dip angles of 20–30°.
The vegetation can be characterized as Mediterranean sclerophyllous forest with *Cryptocarya alba* and *Lithraea caustica* as
dominant plants (Luebert and Pliscoff, 2006; Oeser et al., 2018; Fig. 1b,c). The annual precipitation rate (measured from April
2016 to April 2020) is 346 mm yr$^{-1}$ (Übernickel et al., 2020) and the Holocene net primary production is $280 \pm 50$ g C m$^{-2}$ yr$^{-}$
$^{1}$ (Werner et al., 2018; Oeser and von Blanckenburg, 2020). Records of long-term meteorological data (e.g., precipitation at
ground level, soil water content, air temperature, relative humidity) from a weather station near the study site can be found in
Übernickel et al. (2020).
The regolith profile developed on top of Upper Cretaceous intrusions of mainly granodiorites and tonalites with subordinate
quartz monzodiorites (Gana et al., 1996). The depths of the soil horizons are A: 0–30 cm, B: 30–83 cm and C (saprolite): >83

cm (Fig. 1d). Uplift rates for the north of Santiago de Chile vary between 0.01 and 0.23 mm yr$^{-1}$ with a general mean value of 0.13 ± 0.04 (Melnik, 2016). The soil denudation rate in the nearby La Campana National Park is 53.7 ± 3.4 (S-facing slope) to 69.2 ± 4.6 t km$^{-2}$ yr$^{-1}$ (N-facing slope) (Oeser et al., 2018) or assuming a material density of 2.6 g cm$^{-3}$, 0.024 mm yr$^{-1}$ on average.

## 2.2 Nahuelbuta (NA)

The investigated soil pit (-37.79371° N, -72.95065° E, 1113 m) and the drilling site next to it (-37.79381° N, -72.95043° E, 1114 m) are located approximately 20 km west of Angol (Region IX (Araucanía), Province Malleco) in southern Chile (Fig. 1a). The borehole was located on a plateau-like ridge with gently dipping slopes (ca. 10°).

The pre-land-use vegetation in the study area resembled the recent vegetation found in the Nahuelbuta National Park which can be characterized as temperate forest with *Araucaria araucana* as the dominant tree (Luebert and Pliscoff, 2006; Fig. 1e). However, extensive modern pastoral farming (cow grazing) and fires have converted the ecosystem in the study area to a sparse forest of deciduous trees such as *Nothofagus obliqua* (see Oeser et al., 2018; Fig. 1f). Numerous signs of burning can be observed in the field and charcoal is an integral component of the soil down to 25 cm (A horizon). The precipitation rate (measured from end of March 2016 to April 2020) is 1927 mm yr$^{-1}$ (Übernickel et al., 2020) and the Holocene net primary production is 520 ± 130 g C m$^{-2}$ yr$^{-1}$ (Werner et al., 2018; Oeser and von Blanckenburg, 2020). Records of long-term meteorological data (e.g., precipitation at ground level, soil water content, air temperature, relative humidity) from a weather station near the study site can be found in Übernickel et al. (2020).

The regolith profile developed on top of granitoid rocks of the Nahuelbuta central pluton which contains heterogenous lithological portions (Hervé, 1977; Ferraris, 1979). It is part of the Nahuelbuta Batholith which in turn belongs to the late Carboniferous Chilean Coastal Batholith (Steenken et al., 2016; Deckart et al., 2013). The depths of the soil horizons are A: 0–25 cm, B: 25–90 cm, and C (saprolite): >90 cm (Fig. 1g). Today´s exhumation rates in NA are high (>0.2 mm yr$^{-1}$; Glodny et al., 2008b), whereas the catchment-wide denudation rate is small (27.4 ± 2.4 mm kyr$^{-1}$; van Dongen et al., 2019) compared to LC. The soil denudation rate in the nearby Nahuelbuta National Park ranges between 17.7 ± 1.1 (N-facing slope) to 47.5 ± 3.0 t km$^{-2}$ yr$^{-1}$ (S-facing slope) (Oeser et al., 2018) or assuming a material density of 2.6 g cm$^{-3}$, 0.013 mm yr$^{-1}$ on average. Tectonic fractures in NA can be related to the Lanalhue Fault Zone (see Glodny et al., 2008a).

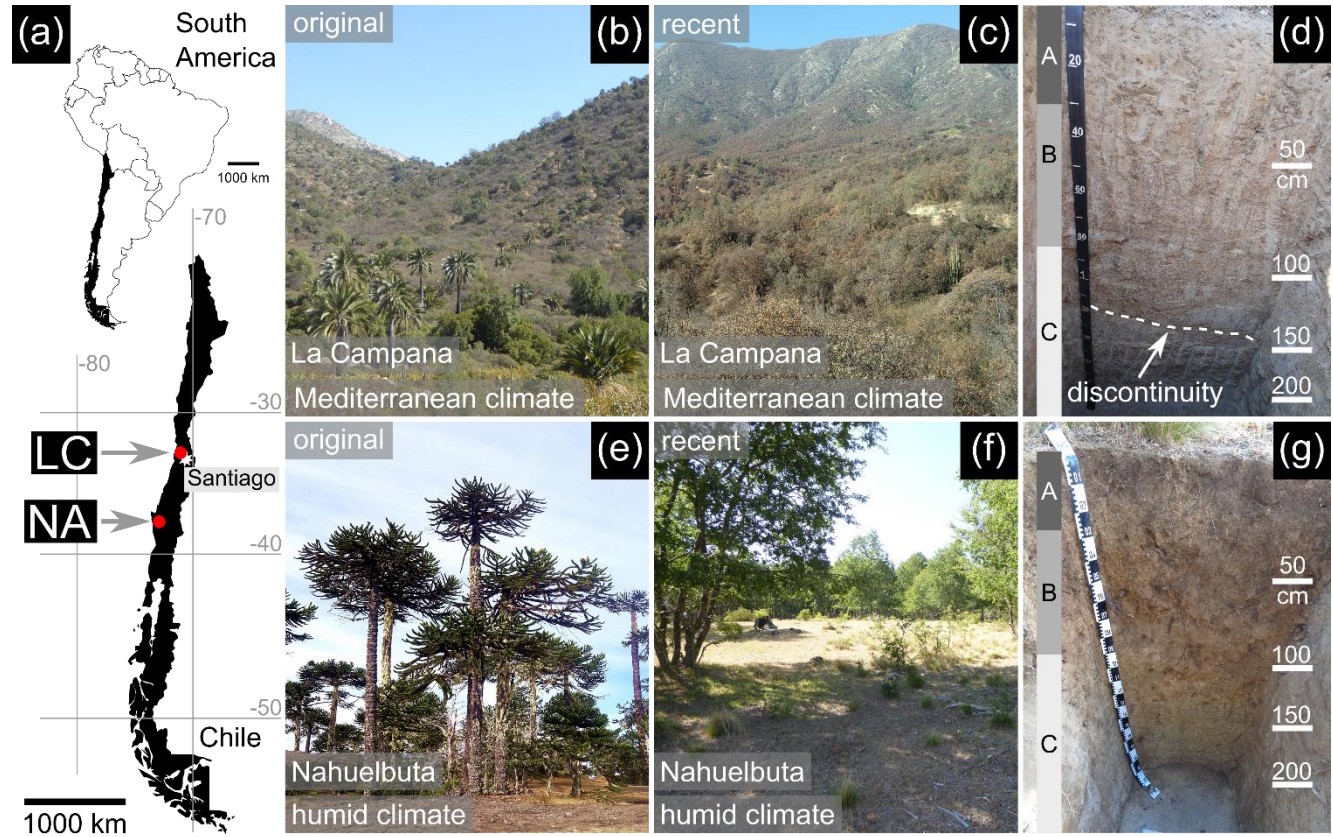

Figure 1: Overview of the study sites and soil profiles. (a) Position of La Campana (LC) and Nahuelbuta (NA) in Chile. Modified map data from **OpenStreetMap** (© **OpenStreetMap** contributors). (b) Original vegetation in LC (i.e., before human intervention; La Campana National Park). (c) Vicinity of the soil pit and drilling site in LC and (d) the first 2 m of the soil profile in LC with inscribed soil horizons (A-C). A prominent discontinuity (dashed line) can be found in the depth interval 120–140 cm. (e) The original vegetation in NA (i.e., before human intervention; Nahuelbuta National Park) in comparison to (f) the recent vegetation in the vicinity of the soil pit and drilling site. (g) The first 2 m of the soil profile in NA with inscribed soil horizons (A-C).

## 3 Materials and methods

### 3.1 Soil pit sampling, drilling, and sample preparation

The sampled 6 m deep soil profiles were located close to the main boreholes at the respective sites. Bulk samples were collected in 20 intervals in each soil pit and weighed around 3 kg. Corestones were not encountered in the soil pit profiles of LC and NA. By using a rotary splitter (type PT, Retsch) the bulk samples were separated into aliquots (see Hampl et al., 2022b). During the drilling campaigns, up to 1.5 m long core runs were recovered by wireline diamond drilling (~80 mm core diameter) using potable water as drilling fluid (see Krone et al. (2021) for a detailed description of the drilling technique). Rock samples were separated from the core by mechanical methods (angle grinder, hammer and chisel), cut (diamond saw), impregnated with blue artificial resin filling the porosity, and subsequently thin-sectioned. Representative bedrock samples were separated from the core (diamond saw) and crushed (jaw crusher).

## 3.2 Analytical methods and calculations

A detailed description of the analytical methods can be found in the accompanying data publication of this study (Hampl et al., 2022b).

### 3.2.1 X-ray fluorescence (XRF) and micro-X-ray fluorescence (μ-XRF)

Soil pit samples were ground with an agate disc mill and annealed (950°C for 1h) before adding a lithium borate flux to produce glass beads in platinum crucibles. The element composition of the glass beads was analysed with a Thermo Scientific ARL PERFORM'X X-ray fluorescence sequential spectrometer (WD-XRF; Thermo Fisher Scientific Inc., U.S.A.). Additional powder pellets were produced by mixing the ground air-dried samples with wax. The mixtures were pressed and analysed with a SPECTRO XEPOS energy dispersive X-ray fluorescence spectrometer (ED-XRF, SPECTRO Analytical Instruments GmbH, Germany). Polished sample slabs of bedrock (Fig. 2) were mapped for the spatial distribution of elements with a μ-XRF spectrometer M4 Tornado (Bruker, Germany).

**Geochemical calculations**

Zr contents obtained from the XRF element analyses on powder pellets were used as an immobile element for the calculation of the chemical depletion fraction (CDF; Riebe et al., 2003; Eq. 1), and the mass transfer coefficient (τ; Anderson et al., 2002; Eq. 2).

$$\text{CDF} = 1 - \frac{Zr_N^b}{Zr_N^w}, \tag{1}$$

$$\tau = \frac{X^w \cdot Zr^b}{X^b \cdot Zr^w} - 1, \tag{2}$$

$X^b$ = concentration of element X in the bedrock, $X^w$ = concentration of element X in the weathered sample, $Zr^b$ = concentration of Zr in the bedrock, $Zr_N^b$ = zirconium content of the bedrock normalized to a LOI-free sum of 100 % (see Hampl et al., 2022b), $Zr^w$ = concentration of Zr in the weathered sample, $Zr_N^w$ = zirconium content of the weathered sample normalized to a LOI-free sum of 100 % (see Hampl et al., 2022b).

The chemical index of alteration (CIA; Nesbitt and Young, 1982) was modified to ΔCIA (Eq. 3).

$$\Delta\text{CIA} = \left[\left(\frac{Al_2O_3^w}{Al_2O_3^w + CaO^w + Na_2O^w + K_2O^w}\right) - \left(\frac{Al_2O_3^b}{Al_2O_3^b + CaO^b + Na_2O^b + K_2O^b}\right)\right] \cdot 100, \tag{3}$$

$w$ = in the weathered sample, $b$ = in the bedrock.


### 3.2.2 Oxalate- and dithionite extraction

Air-dried bulk samples of <2 mm (dry-sieved) were used for oxalate- and dithionite extractions. The solutions thus obtained
were measured with an ICP-OES iCAP 6300 DUO (Thermo Fisher Scientific, USA) to determine the extractable Fe, Al, and
Si contents. The oxalate extraction employed targets the easily extractable mainly X-ray amorphous Fe(III) oxyhydroxides and
(poorly) crystalline Al-containing minerals (see review by Rennert (2019) and references therein). The dithionite extraction
dissolves crystalline and amorphous iron oxides (McKeague and Day, 1966). In doing so it can (partly) attack Al-bearing
(mineral) phases (see review by Rennert (2019) and references therein).
The oxalate extractions were performed after Schwertmann (1964) with an oxalic acid-/oxalate-extraction solution (0.2 M, pH
3.0). After the addition of the solution to the sample and shaking for 2 h in the dark (over-head shaker), the solution was filtered
in a darkened room and immediately measured. The cold dithionite extractions were performed based on Holmgren (1967)
with an extraction solution (mixture of 0.2 M $NaHCO_3$ and 0.24 M trisodium citrate) and sodium dithionite under oxic
conditions. The resulting mixture of chemicals and sample was shaken for 16 h and centrifuged before the supernatant was
filtered and immediately measured. Additional reference samples, blanks and calibration solutions were also prepared and
measured like the soil pit samples. The results of the samples presented here are the mean of duplicate measurements performed
on two individually extracted sample aliquots.

### 3.2.3 Grain size determination

Sample aliquots were suspended in de-ionized water ($<10$ $\mu$S m$^{-1}$) and dispersed in a rotating overhead shaker (approx. 15 h)
and a subsequent ultrasonic bath before vibrational wet sieving. The >63 $\mu$m sieving fractions were dried (50°C, approx. 24
h) and their weight percentages were measured. The clay and silt contents were determined using the <63 $\mu$m suspension and
a pipette method. Organic-rich samples were treated with $H_2O_2$ to decompose organic matter and sodium pyrophosphate was
used as a dispersion agent to prevent coagulation. Clay (<2 $\mu$m) was separated from the <63 $\mu$m fraction slurry via
centrifugation.

### 3.2.4 X-ray diffraction (XRD)

Untreated air-dried aliquots of bulk samples were crushed in a porcelain mortar and afterwards processed with a micronisation
XRD-mill McCrone (Retsch, Germany) to obtain a final powder of <10 $\mu$m. These powders were mounted to XRD sample
holders by back-loading and X-ray diffraction measurements were performed with a Rigaku SmartLab equipped with a 9 kW
rotating Cu-anode and a HyPix-3000 detector in Bragg-Brentano geometry (3–80° 2θ, scan step: 0.01°, scan speed: 1° min$^{-1}$,
and 60 rpm sample rotation). For the identification and semi-quantitative analyses, the software SmartLab Studio II and the
mineral database PDF-4 Minerals 2021 including reference intensity ratio (RIR) factors were used. Image processing (imageJ;
version 1.53a; Schneider et al., 2012) performed on the µ-XRF element distribution maps in Fig. 2 was used to get a rough

compositional information of the mineral content in the sampled bedrock. These analyses were used as a supporting basis for the semi-quantitative XRD analyses with RIR factors.

Clay mineral contents in the samples were quantitatively estimated by combining the results of the grain size determination with the semi-quantitative results of the XRD analyses. The clay-size fraction (<2 µm) of which the mass was determined by sieving/pipetting, was assumed to represent the entire clay mineral content of the sample, while the other size fractions were considered to be free of clay minerals. This assumed clay mineral content (in wt.%) was combined with the XRD-semi-quantitative weight percentages of the primary minerals in the same sample to approximate the mineral composition of the whole soil pit sample (summarized to 100 wt.%). Despite the assumption that only the <2 µm grain size fraction contains clay minerals, this estimate appears to be the most accurate because there are no matching files in the mineral database used here that would accurately semi-quantify the identified interstratified clay minerals.

The separated clay-size fractions were measured as oriented clay films (texture preparation). A D2 Phaser XRD device (Bruker) equipped with a Cu-anode was utilized for the measurements. The diffractograms were recorded in Bragg-Brentano geometry in the range of 3–35° 2θ (step width: 0.01° 2θ, 0.5 seconds per step). The samples were measured after air-drying, during ethylene glycol saturation and after a thermal treatment at 550°C for 1 h. Selected samples were also treated with glycerol and KCl (1 M) to characterize the clay minerals in more detail. The identification was supported by a clay mineral identification chart (Starkey et al., 1984).

**3.2.5 Magnetic susceptibility measurements**

The magnetic susceptibility was measured on all twenty-one McCrone-milled bulk samples of the LC profile with a KLY-3 Kappabridge (AGICO, Czechia). Measurements were performed in triplicates at room temperature, a frequency of 875 Hz and a peak magnetic field of 300 A m$^{-1}$.

To obtain the magnetite content of the bedrock, a representative 60x60 mm sample slab (Fig. 2a) was mapped with the µ-XRF spectrometer M4 Tornado. The µ-XRF map that depicts only the maximum Fe content was used as an approximation of the magnetite content since magnetite is the mineral with the highest Fe concentration in the rock. Finally, the map was analysed with the image processing program imageJ (version 1.53a; Schneider et al., 2012) to quantify the magnetite content. The obtained value was equalled to the measured magnetic susceptibility of the same sample and used to convert the magnetic susceptibility results of the LC soil pit samples into approximated magnetite contents by the rule of three. The investigated bedrock of NA contains no magnetite.

**3.2.6 Light microscopy and electron microprobe analysis (EMPA)**

Thin sections were investigated with the light microscope DM750P (Leica, Wetzlar, Germany) equipped with a microscope camera (Euromex, The Netherlands). Electron microprobe element distribution maps of selected areas were obtained for Al, Ca, Fe, K, and Mg by using standard wavelength dispersive techniques on a JEOL Superprobe JXA-8230 fitted with a W-emitter electron gun (accelerating voltage: 15 kV, beam current: 20 nA, beam diameter and step width: 1 µm).

## 4 Results

The data tables (cited as Table S1–S5) are included in the accompanying data publication (Hampl et al., 2022b).

### 4.1 Bedrock

According to the Streckeisen nomenclature the bedrock of LC can be described as granodiorite and the investigated bedrock of NA can be described as granite. However, the drill core revealed that the bedrock of NA occasionally contains more mafic sections. The most abundant minerals in the fine-grained bedrock of LC are plagioclase, quartz, microcline, hornblende, biotite, and chlorite (Fig. 2a,b). The latter occurs solely and abundantly along with (former) biotite crystals as their hydrothermal transformation products (i.e., chloritization; e.g., Kogure and Banfield, 2000). Magnetite is a ubiquitous accessory mineral (Fig. 2c; <1 vol.%) in LC and shows no signs of alteration to hematite (martitisation). Pyrite and chalcopyrite are also observed in much smaller abundance than magnetite. Mafic xenoliths can frequently be found in the granodiorite of LC.

In the coarse-grained Nahuelbuta granite, quartz, plagioclase, microcline, biotite, and chlorite are the main constituents (Fig. 2d,e). In contrast to LC, amphiboles can only be found as an accessory mineral (<1 vol.%) in the investigated bedrock of NA. Like in LC, biotite is often chloritized. Magnetite and sulfides could not be identified in the investigated rock samples of NA. Variations in the biotite content, the occurrence of amphibole crystals, differences in fabric (microcline of a few centimetres), the alternation with mafic portions and the presence of pegmatites in the core make the overall lithology of NA far more heterogenous compared to LC.

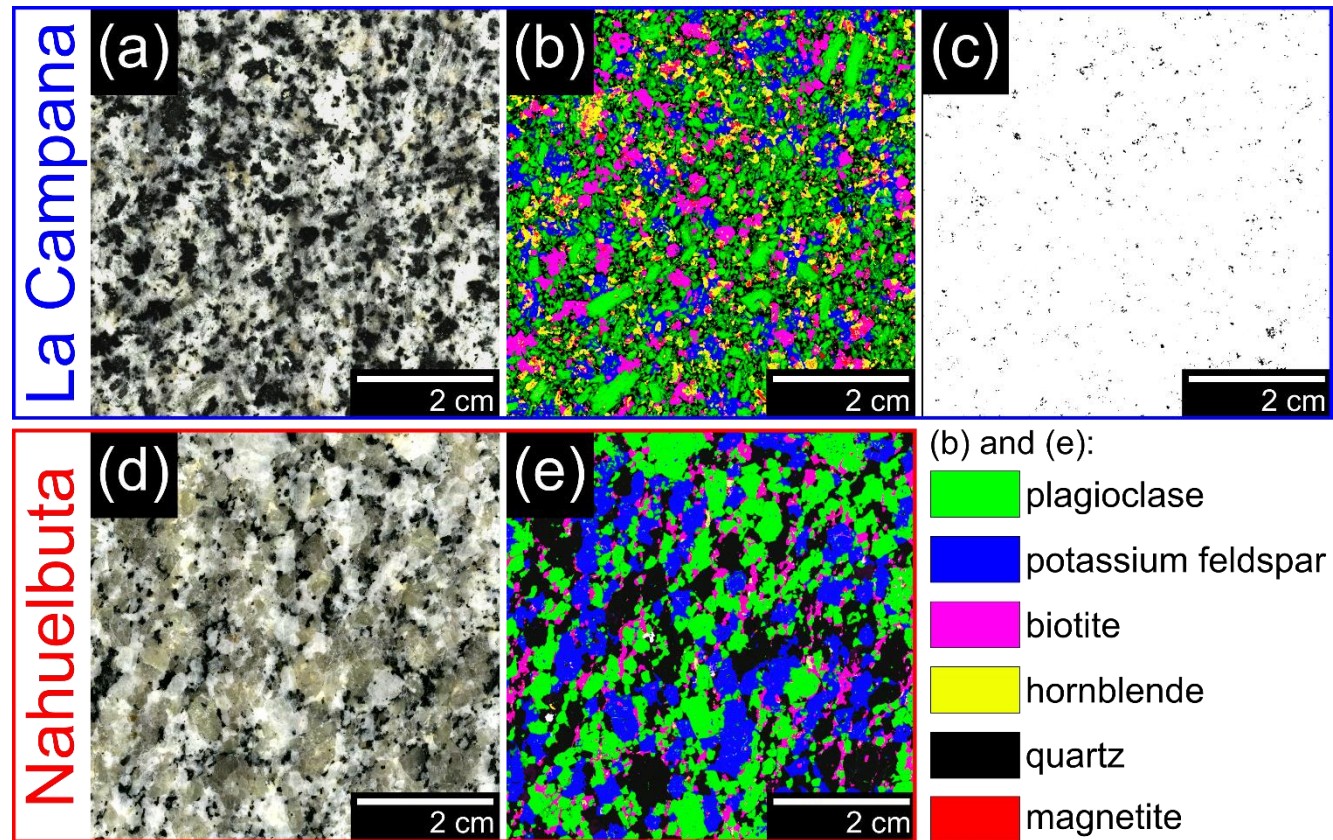

Figure 2: Bedrock of the investigated profiles. (a) Bedrock from La Campana (IGSN: GFFJH0095) with (b) a corresponding μ-XRF map reflecting the spatial mineral distribution. (c) μ-XRF map of the maximum Fe content (black dots) representing the magnetite crystals in the bedrock sample slab of La Campana. (d) Typical unweathered granite from Nahuelbuta (IGSN: GFFJH00H0) and (e) a μ-XRF map reflecting the mineral content of the same.

## 4.2 Regolith

### 4.2.1 Incipient weathering in rock

Weathered rock from the borehole of LC shows abundant indications of weathering-induced fracturing (WIF) due to Fe(II) oxidation in biotite, like fanned-out edges or opening due to dilatation (Fig. 3a,b). Secondary minerals like Fe(III) oxyhydroxides are subordinate and are mostly associated with biotite. They are detectable as Fe-enrichments at the edge of biotite crystals and within the cracks encompassing biotite (Fig. 3c,d). To a minor degree, Fe(III) oxyhydroxides are also associated with hornblende. Nevertheless, most micro-fractures in feldspar and quartz of the investigated thin sections are solely filled with blue resin and are bare of any secondary minerals.

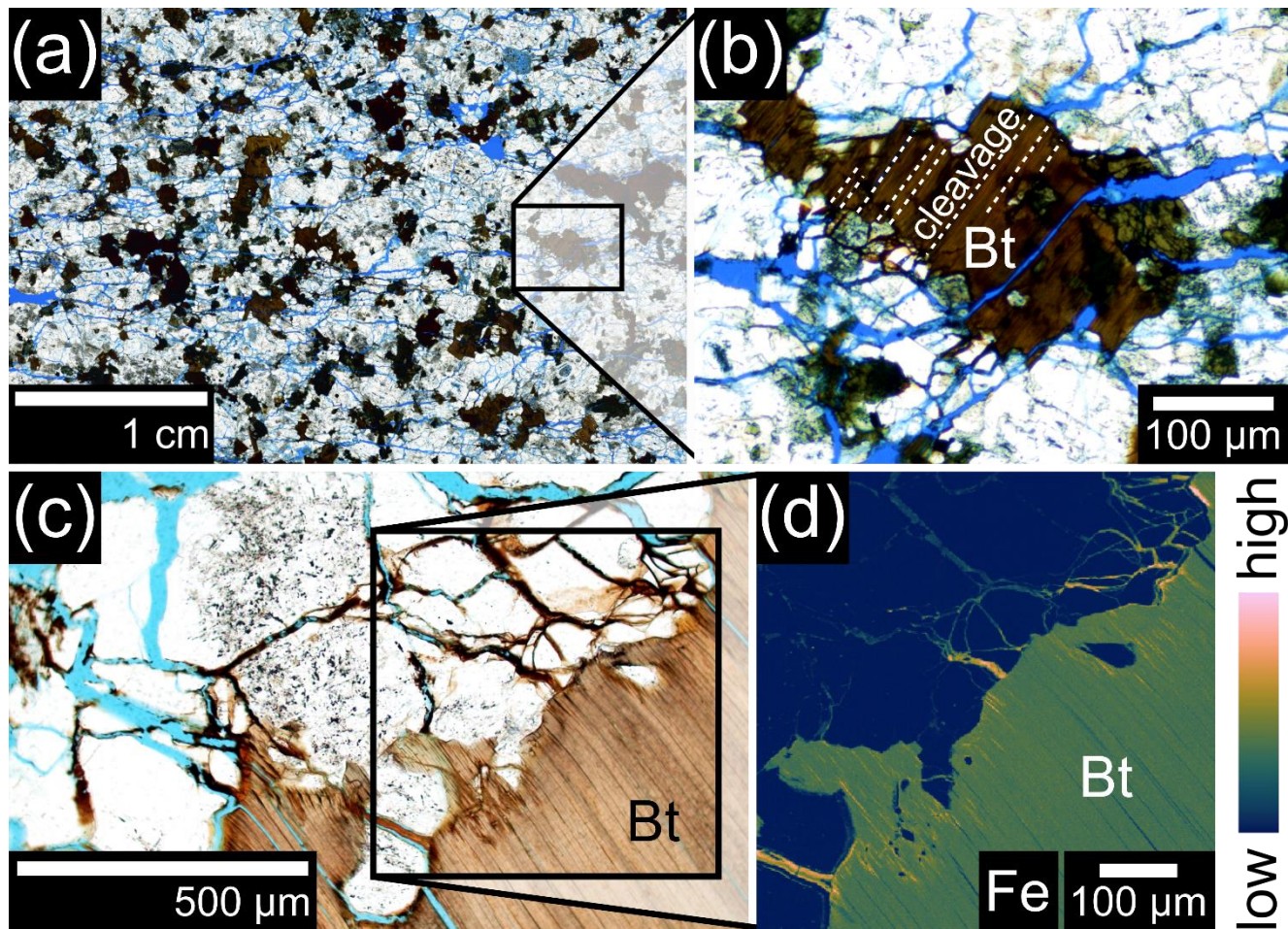

Figure 3: Rock weathering in La Campana (LC; porosity is represented by blue-dyed resin). (a) Thin section image (transmitted light) of a weathered rock sample obtained from approx. 27 m depth in the LC drill core (IGSN: GFFJH00HY). (b) A detail image of biotite showing signs of dilatation (dashed lines indicate cleavage planes). (c) Secondary minerals in cracks around biotite. (d) The electron microprobe map of the contact zone between biotite and quartz/feldspar displays Fe-enrichments at the interface. Bt = biotite.

Indications of WIF around biotite are also present in weathered rock of NA (Fig. 4a,b). However, the cracks are often filled and covered with Fe(III) oxyhydroxides and clay minerals as observed with light microscopy (Fig. 4c) and electron microprobe investigations. Unlike LC, weathered rock in NA is characterized by distinct Ca-depletion and Al-enrichment in plagioclase which indicates partial dissolution (Fig. 4d–f). These alteration sites host secondary minerals covering the newly formed surfaces which were formed by the dissolution of the plagioclase.

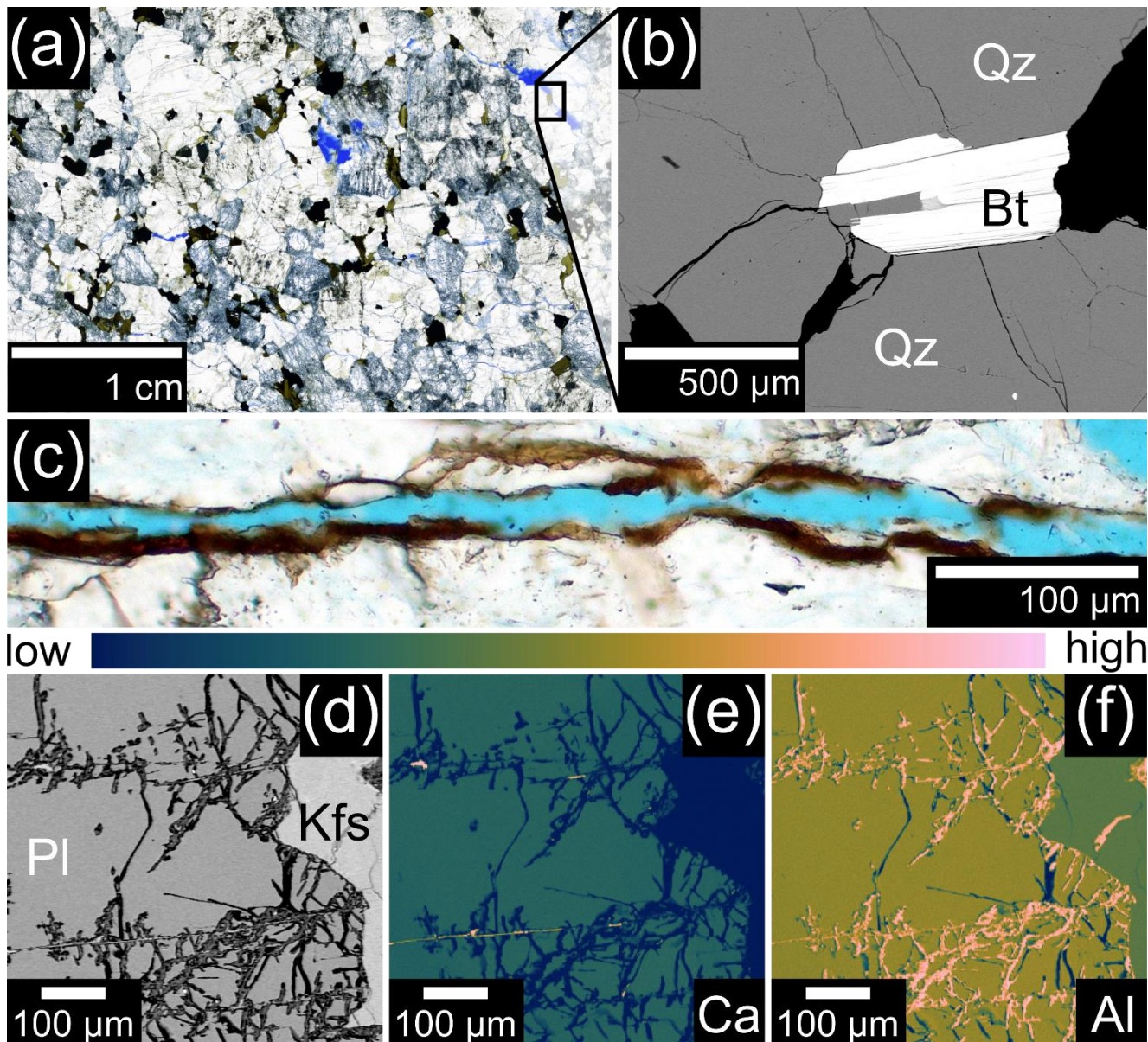

**Figure 4: Rock weathering in Nahuelbuta (NA). (a) Thin section image (transmitted light) of weathered rock obtained from approx. 6 m depth in the NA drill core (note that the porosity (blue) is largely associated with weathered plagioclase; IGSN: GFFJH00HX). (b) Indications of WIF in quartz (backscattered electron image, EMP). (c) Thin section image (transmitted light) of a crack covered with brown Fe(III) oxyhydroxides from approx. 12 m depth (IGSN: GFFJH00J2). (d) Backscattered electron image (EMP) of partly dissolved plagioclase and (e) the respective Ca and (f) Al map of the section (IGSN: GFFJH00HX). Qz = quartz, Bt = biotite, Pl = plagioclase, Kfs = potassium feldspar.**

### 4.2.2 Saprolite and soil

**Chemical alteration**

The mass transfer coefficient $\tau$ indicates moderate depletion below 80 cm (not smaller than -0.2) in the LC soil pit profile, but clear depletion in the uppermost few decimetres where Na, K, Mg, Ca, Si and P can reach up to $\tau = $ -0.5 and -0.6 (Fig. 5; Table S1). A pronounced P depletion can be detected down to 1.4 m depth in LC. The chemical depletion fraction (CDF) of LC and the bedrock-normalized chemical index of alteration ($\Delta$CIA) indicate a weak chemical weathering degree below ca. 0.5–1 m, but minor chemical depletion was analysed down to the bottom of the 6 m deep profile of LC (see $\Delta$CIA; Fig. 5). In contrast, Nahuelbuta is characterized by distinct chemical depletion of Ca and Na (up to $\tau = $ -0.9; Fig. 5). K is depleted to a depth of approximately 5 m, Si to a depth of ~6 m and Mg shows moderate depletion ($\tau \geq$ -0.3) throughout the profile. P is strongly depleted between ca. 2–6 m ($\tau \sim$ -0.6) but the P content gradually increases from approx. 3 m depth towards the surface and is enriched in the uppermost ~20 cm of the soil (A horizon; Fig. 5). The CDF values of NA indicate depletion down to the bottom of the profile at 6 m depth. The $\Delta$CIA of the profile underpins strong chemical alteration compared to the bedrock (Fig. 5). However, overall chemical depletion decreases towards the bottom of the soil profile and according to the $\tau$-values in 550–600 cm only Na, Ca and P seem to be significantly depleted at >6 m depth.

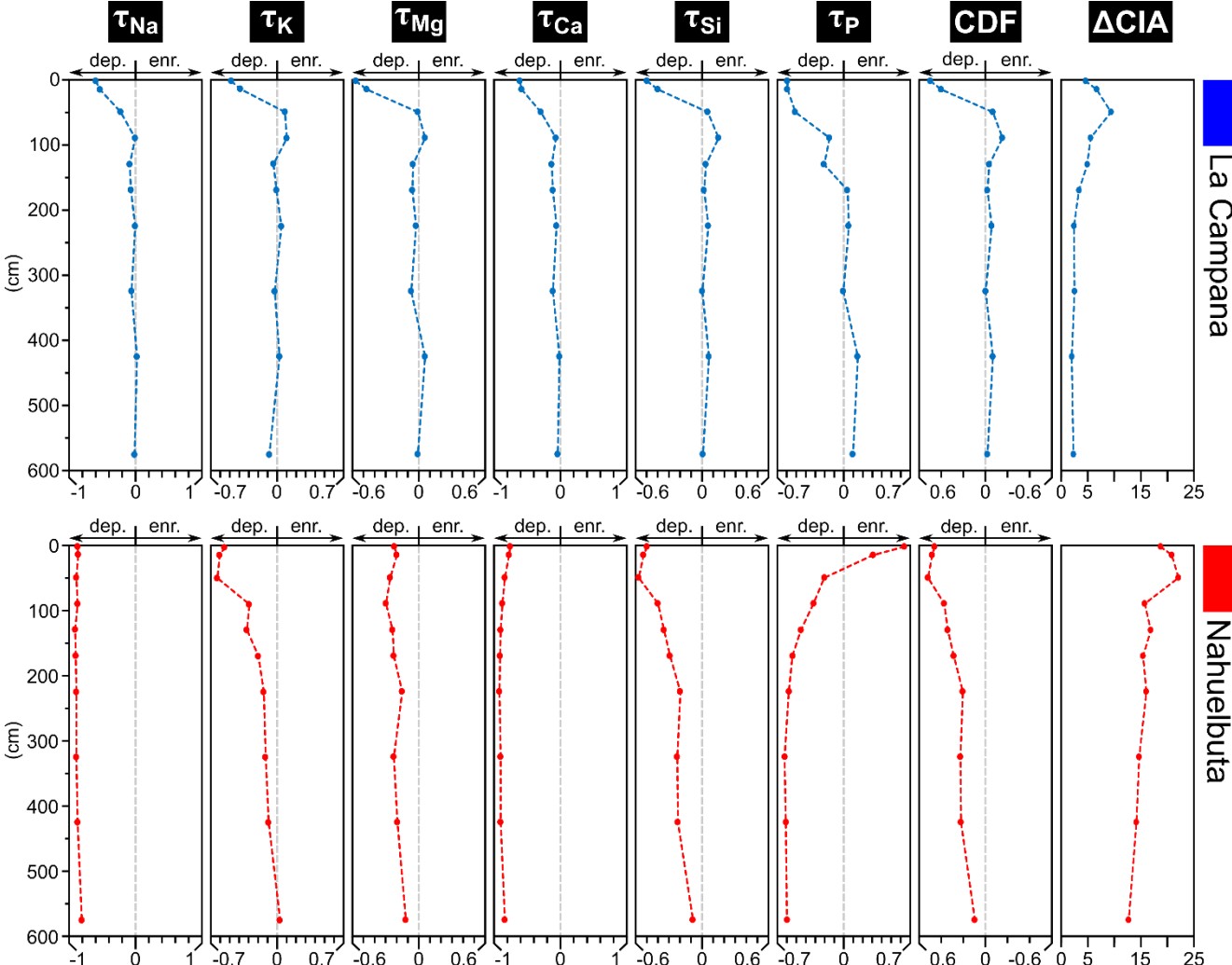

Figure 5: τ-values of Na, K, Mg, Ca, Si, and P as well as the CDF (all based on Zr) and ΔCIA values of the soil pit profiles in La Campana (LC) and Nahuelbuta (NA; Table S1). Note that the scales are equal for the individual indices of LC and NA. dep. = depletion, enr. = enrichment.

Since many secondary minerals are formed via a metastable or amorphous precursor (e.g., Steefel and van Cappellen, 1990; Hellmann et al., 2012; Behrens et al., 2021), we assume that the extractable Fe, Si, and Al contents are indicative for recent weathering of primary minerals (Fig. 6; see chapter 3.2.2 for an assignment of the extractable elements to the minerals they likely originate from).

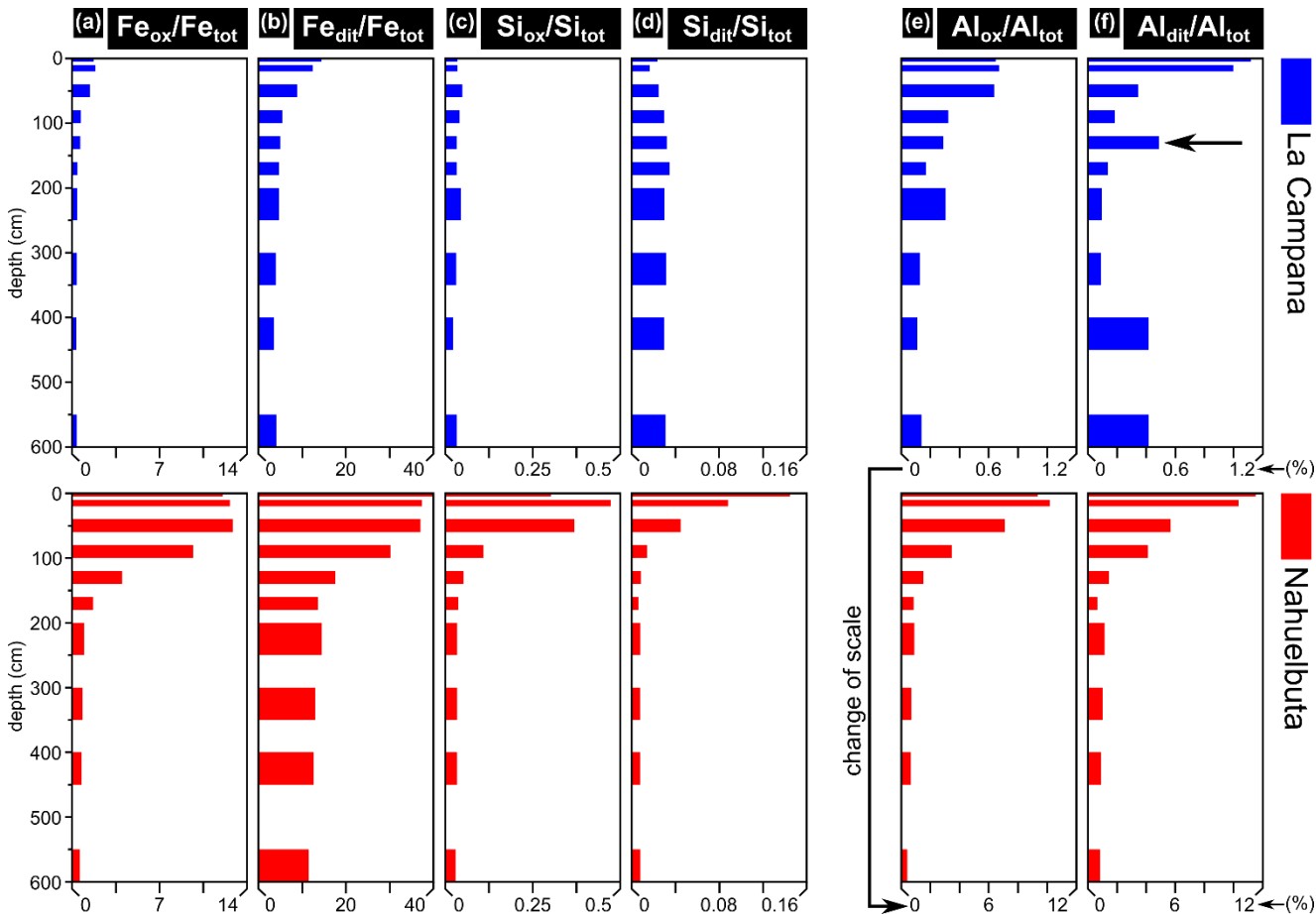

**Figure 6: Oxalate- and dithionite-extractable Fe, Si and Al contents divided by the respective total element contents of the bulk soil**
**pit samples of La Campana (LC) and Nahuelbuta (NA; Table S2). The elevated ratio at 120–140 cm in (f) (arrow) coincides with the**
**position of a discontinuity in the profile (Fig. 1d). Note that the scales for LC and NA are equal in (a)–(d). The scale in (e) and (f) is**
**one order of magnitude larger for NA compared to LC.**

Extractable contents of Fe in LC are moderately elevated in the uppermost meter of the profile (up to $Fe_{dit}/Fe_{tot}$ ~14 %)
compared to the other depth intervals which show low contents ($Fe_{dit}/Fe_{tot}$ <1 m: ~4–5 %; Table S2; Fig. 6a,b). The extractable
Si-contents show no clear pattern (Fig. 6c,d) whereas oxalate-/dithionite-extractable Al-contents are variable in the profile of
LC (Fig. 6e,f). The elevated $Al_{did}/Al_{tot}$ value in the depth interval 120–140 cm in LC (~0.5 %; Fig. 6f) coincides with a
discontinuity in the saprolite (Fig. 1d) and may indicate more secondary crystalline and amorphous Al-bearing phases in this
section. The profile in NA is characterized by high amounts of extractable Fe, Si and Al contents which are especially elevated
in the uppermost meter of the profile ($Fe_{dit}/Fe_{tot}$ up to ~40 %, $Si_{dit}/Si_{tot}$ up to ~0.14 %, $Al_{dit}/Al_{tot}$ up to ~12 %). The extractable
contents rapidly decrease from the surface towards the bottom of the NA profile and starting at approx. 2 m they are similar
down to 6 m (Fig. 6).
The $Fe_2O_3$ content in the investigated bedrock of LC is more than twice as high as that of the NA bedrock, but the oxalate- and
dithionite-extractable Fe contents (and hence the amount of the respective secondary minerals) are far higher in NA (Fig. 6a,b).
The difference between LC and NA is even more pronounced for the extractable Al contents as values in NA can be 10 times
higher than in LC (Fig. 6e,f). The extractable contents in the profiles of both study sites are generally within the range of
previous investigations on soil samples from the La Campana and Nahuelbuta National Parks, but the $Fe_{dit}/Fe_{tot}$ contents in the
upper profile section of NA in this study are much higher (up to 40 %) than those measured in the Nahuelbuta National Park
(<25 %; Oeser et al., 2018).

**Mineral content and grain sizes**

The sieving results of LC show a gradual decrease in particle size from the bottom of the profile towards the surface and a
relatively constant sand-size content ranging from 65–80 wt.% with similar portions of the individual sand-size fractions (Fig.
7a). The small geochemical depletion below the uppermost ~2 m of the LC profile (Fig. 5) is also reflected in the little changing
mineral composition of the profile (Fig. 7b). Only the plagioclase (Ca-albite) content slightly decreases from approx. 1 m depth
towards the surface. A small decrease of biotite in the depth interval 120–140 cm coincides with the mentioned discontinuity
of this profile section (Fig. 1d). The abundant chlorite of the investigated bedrock in LC (~5 wt.%) is completely weathered
and absent from the soil pit samples (Fig. 7b).
Significant alteration of magnetite (e.g., martitisation) could not be observed in ore microscopic investigations of the magnetic
particles in soil pit samples of LC. Thus, the magnetic susceptibility directly reflects the magnetite content of the samples (e.g.,
Ferré et al., 2012). A relative magnetite enrichment was detected in the uppermost 40 cm of the LC profile (1–1.6 vol.%)
whereas the rest of the profile shows approximately constant magnetite contents (mean ~0.9 vol.%) close to the value of the
investigated bedrock (0.94 vol.%; Fig. 7c). This almost consistent magnetite content underlines the homogeneity of the bedrock
that was weathered in the 6 m deep soil pit (i.e., no mafic dykes, pegmatites, or major xenoliths).
The soil pit profile of NA is characterized by a much higher gravel- and silt/clay-size content compared to LC (Fig. 7d). This
reflects the more heterogeneous grain size distribution of the investigated bedrock in NA compared to the bedrock of LC (see
Fig. 2). High clay contents can be detected in the uppermost meter of the NA profile (partly >20 wt.%) and the identified
mineral content of the soil pit samples differs significantly from the mineral content of the investigated bedrock (Fig. 7e). The
plagioclase (Ca-albite) content distinctly decreases from the bottom of the profile towards the surface and the bedrock content
of ~28 wt.% partly decreases down to 1 wt.% in the soil pit. The microcline content on the other hand is relatively uniform.
Just as in LC, the chlorite of the bedrock analysed here (~1 wt.%) is completely weathered in the NA soil pit profile and is
absent from the samples.

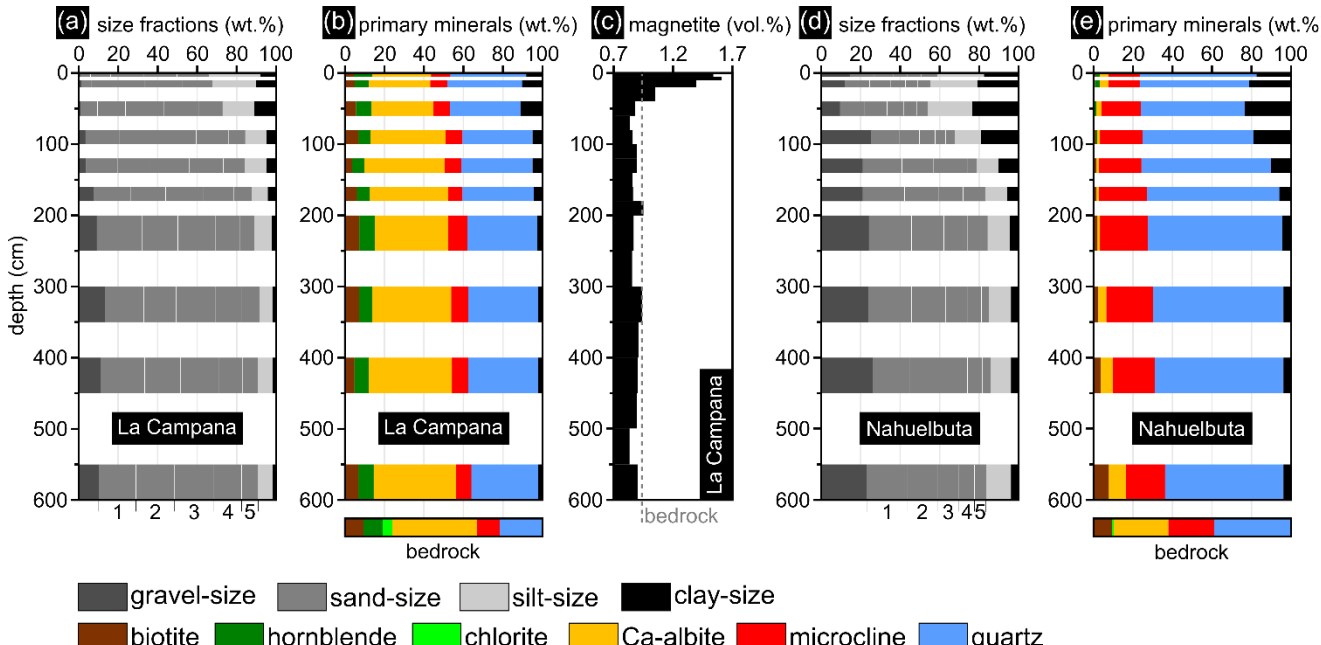

**Figure 7: Sieving/pipetting results, semi-quantitative XRD results and approximated magnetite contents of the investigated soil pit**
**samples in La Campana (LC) and Nahuelbuta (NA). (a) Grain size distribution based on wet-sieving and pipetting, (b) semi-**
**quantitative XRD and (c) magnetic susceptibility results converted to approximate magnetite contents of the LC profile. (d) Wet-**
**sieving combined with pipetting results and (e) semi-quantitative XRD results of the NA samples. Semi-quantitative XRD results of**
**the investigated bedrock samples (see Fig. 2) are given below the results of the soil pit samples. In (a) and (d): 1: ≤2000 to >1000 µm,**
**2: ≤1000 to >500 µm, 3: ≤500 to >250 µm, 4: ≤250 to >125 µm, 5: ≤125 to >63 µm.**

The mineral content of the clay-size fraction in LC differs significantly from that in NA (Fig. 8). La Campana is characterized
by abundant expandable clay minerals (interstratified chlorite-smectite and interstratified mica-smectite) which can largely be
traced back to the weathering of chlorite and biotite (Fig. 8a). Kaolinite can be found throughout the LC profile whereas
interstratified mica-vermiculite only occurs in the depth interval of the discontinuity (120–140 cm; see Fig. 1d). The
expandable portion of the interstratified minerals gradually decreases from the profile bottom towards the surface and cannot
be detected in the uppermost centimetres of the LC profile. Only mica and kaolinite constitute the clay-size fraction of the
uppermost part of the profile in LC. The mineral content in the clay-size fraction of NA is characterized by small amounts of
interstratified mica-vermiculite below 1 m depth and ubiquitous kaolinite which shows small expandable portions below 2 m
depth. Hydroxy-interlayered vermiculite (HIV) and gibbsite can first be detected in 400–450 cm depth and the content
increases towards the surface. The main minerals of the clay-size fraction in the uppermost part of the profile are HIV, kaolinite,
and gibbsite (Fig. 8b).

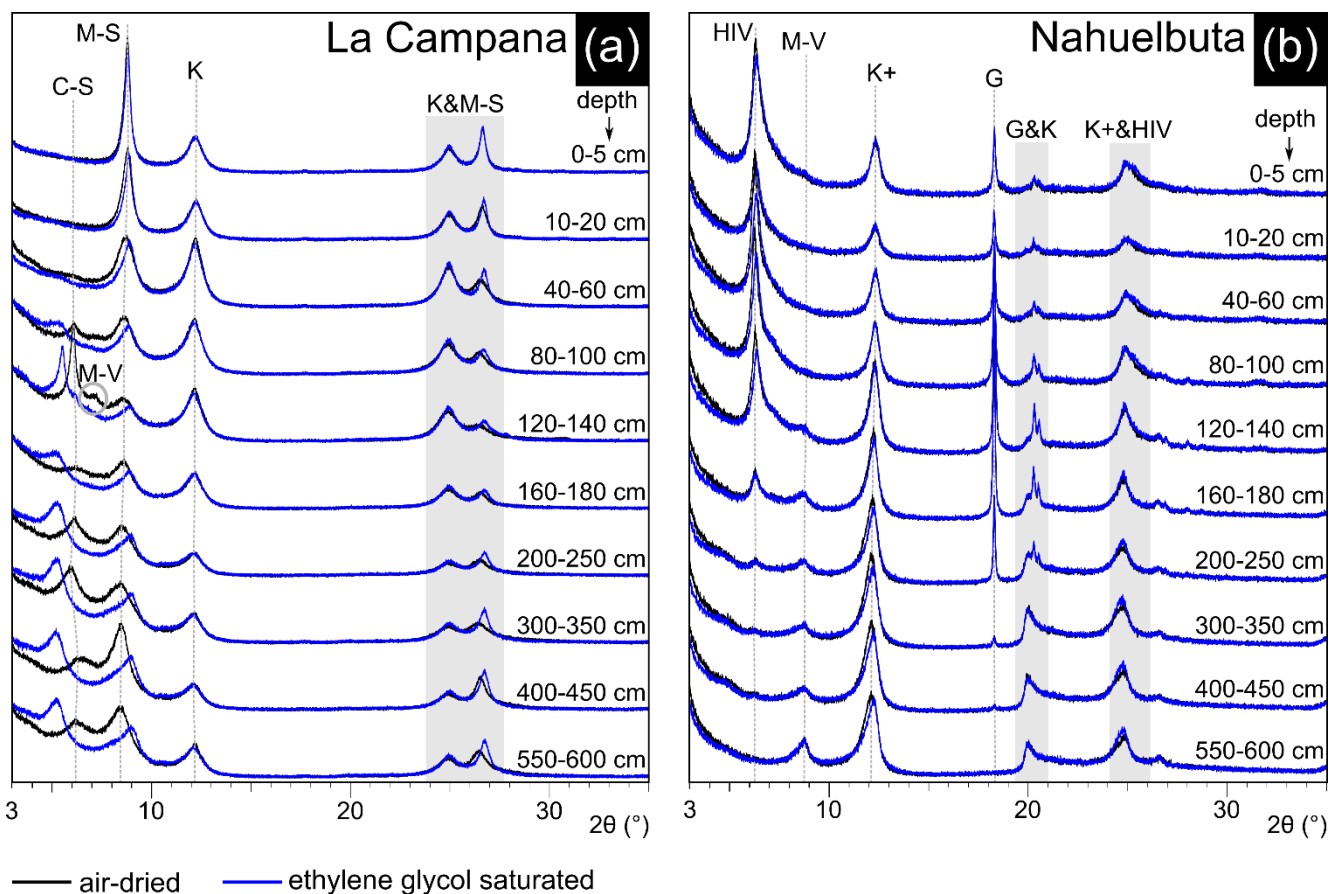

**Figure 8: Minerals in the clay-size fraction of the soil pit profiles in La Campana (LC) and Nahuelbuta (NA). (a) The profile in LC features abundant expandable clay minerals. (b) NA is characterized by the presence of gibbsite and vermiculite but very minor amounts of expandable clay minerals. C-S = interstratified chlorite-smectite, G = gibbsite, HIV = hydroxy-interlayered vermiculite, K = kaolinite, K+ = kaolinite with expandable portions, M-S = interstratified mica-smectite, M-V = interstratified mica-vermiculite.**

## 5 Discussion

### 5.1 Climate-dependent mineral transformations

Chemical depletion and mineral transformations are far more pronounced in the profile of NA compared to the profile of LC even though the bedrock of LC contains more minerals with higher solubility compared to NA (more plagioclase, biotite, chlorite or hornblende in LC than in NA where quartz and potassium feldspar dominate; see e.g., Wilson, 2004; Bandstra et al., 2008). The high chemical depletion ($\tau$[Na, Ca] up to -0.9 and $\Delta$CIA up to 22; Fig. 5) and the occurrence of gibbsite in NA are indicative of distinct dissolution of primary minerals (esp. plagioclase; Fig. 7e) and solute removal of alkali and alkaline earth metals while immobile Al remains as hydroxide (Al(OH)$_3$ = gibbsite). This depletion is assumed to be the result of more water infiltration into the subsurface of NA (more precipitation due to humid climate) compared to LC (less precipitation due to Mediterranean climate). The measured $\tau$[P] distribution in NA is a clear indication for biologically controlled nutrient uplift

and recycling within the topsoil (Jobbágy and Jackson, 2004). We assume that the high precipitation rate in NA leads to more
biomass production by plants, which in turn implies more litter production and a stimulation of biogenic decay that supplies
plants with nutrients. Thus, we concur with the hypothesis that the ecosystem in NA is thriving on nutrient recycling rather
than on an uptake of nutrients that were released by biogenic weathering at depth (Oeser and von Blanckenburg, 2020). Apart
from Ca, Na, and P ($\tau$-values in Fig. 5), the chemical depletion successively decreases from the surface towards the bottom
part of the investigated profile in NA. To account for this shallow chemical depletion, we propose that a secondary-mineral-
controlled impeding of the fluid infiltration to depth is playing an important role for the depth of mineral transformations in
NA.
Chemical depletion can be detected throughout the investigated profile in LC, but the chemical weathering degree is very low
between 2–6 m depth (Fig. 5) and the mineral transformations in this section of the profile are only minor (Fig. 7b). On the
other hand, distinct mineral dissolution and removal of solutes is testified by the higher magnetic susceptibility values in the
uppermost decimetres of the LC profile. This can be related to a residual accumulation of weathering-resistant magnetite while
other minerals like plagioclase dissolve. The strong chemical depletion in this part of the profile is also reflected by the low $\tau$-
and elevated CDF-/$\Delta$CIA-values. To account for the detected weak but deep chemical weathering in LC, we propose that a
secondary-mineral-controlled formation of fluid pathways is facilitating the fluid infiltration to depth and is thus an important
control on the chemical weathering reactions in the subsurface.
The difference between the profiles is also displayed by the oxalate- and dithionite-extractable Fe, Si, and Al contents. While
high extractable contents especially within the uppermost 2 m of the NA profile are interpreted to indicate considerable
ongoing (recent) transformations of primary to secondary minerals, LC shows comparatively little indications in this regard.
This difference underlines the higher degree of mineral transformations in NA compared to LC which is also reflected in the
mineral content of the clay-size fraction (see Fig. 9). That oxalate- frequently exceeds dithionite-extractable Al contents is
indicative for amorphous phases since oxalate is more effective at extracting amorphous forms of Al (McKeague and Day,
1966). Moreover, the highest contents of the clay-size fraction in the profiles are in good correlation with the elevated
extractable Fe, Si and Al contents and highlight the pronounced mineral transformation in the uppermost part of the profiles.
This size fraction hosts most of the products of primary silicate weathering. Clay-size minerals of NA mainly correspond to
distinct weathering of plagioclase and biotite, whereas in LC they can mainly be associated with chlorite and biotite weathering
(Fig. 9). Feldspar weathers to kaolinite and gibbsite in NA and biotite weathers to hydroxy-interlayered vermiculite (HIV).
Chlorite completely dissolved in the NA profile, whereas both chlorite and biotite in LC weather via interstratified clay
minerals to smectite. Finally, smectite and feldspar likely weather to kaolinite in LC (Fig. 9). The mineral composition of the
clay-size fraction (Fig. 8) is dependent on the bedrock composition (e.g., more chlorite in LC) and the climate-dependent
mineral dissolution (see Fig. 9) in the study sites. However, we argue that the amount of secondary minerals is largely a
function of the climatic conditions that control the weathering intensity via water availability in the study sites.
As this study does not consider the entire weathering profile in LC and NA, the interplay between erosion rate and weathering
advance rate (Lebedeva and Brantley, 2020) is not addressed here. However, the different denudation rates in the study areas
(mean soil denudation rate in LC: ~ 61, in NA: ~ 33 t km$^{-2}$ yr$^{-1}$; Oeser et al., 2018) likely affect the weathering intensity. Due
to the higher denudation rates in LC compared to NA (Oeser et al., 2018; van Dongen et al., 2019), we hypothesize that the
residence time of weathered material in the regolith of LC is shorter than in NA. Thus, there is less time for chemical weathering
in LC. In combination with the lower water availability in LC, this factor might contribute to the lower weathering intensity
in the regolith of LC compared to NA. This would also concur with the finding that water availability in the soil and soil
residence time are the limiting factors for weathering processes in dry environments (Schoonejans et al., 2016). NA, on the
other hand, is characterized by a longer residence time of weathered material. Together with the higher water availability, this
factor might contribute to the high weathering intensity in the upper regolith of NA. The situation in LC may be comparable
to an incompletely developed profile, and the situation in NA to a completely developed profile (Reis and Brantley, 2019).

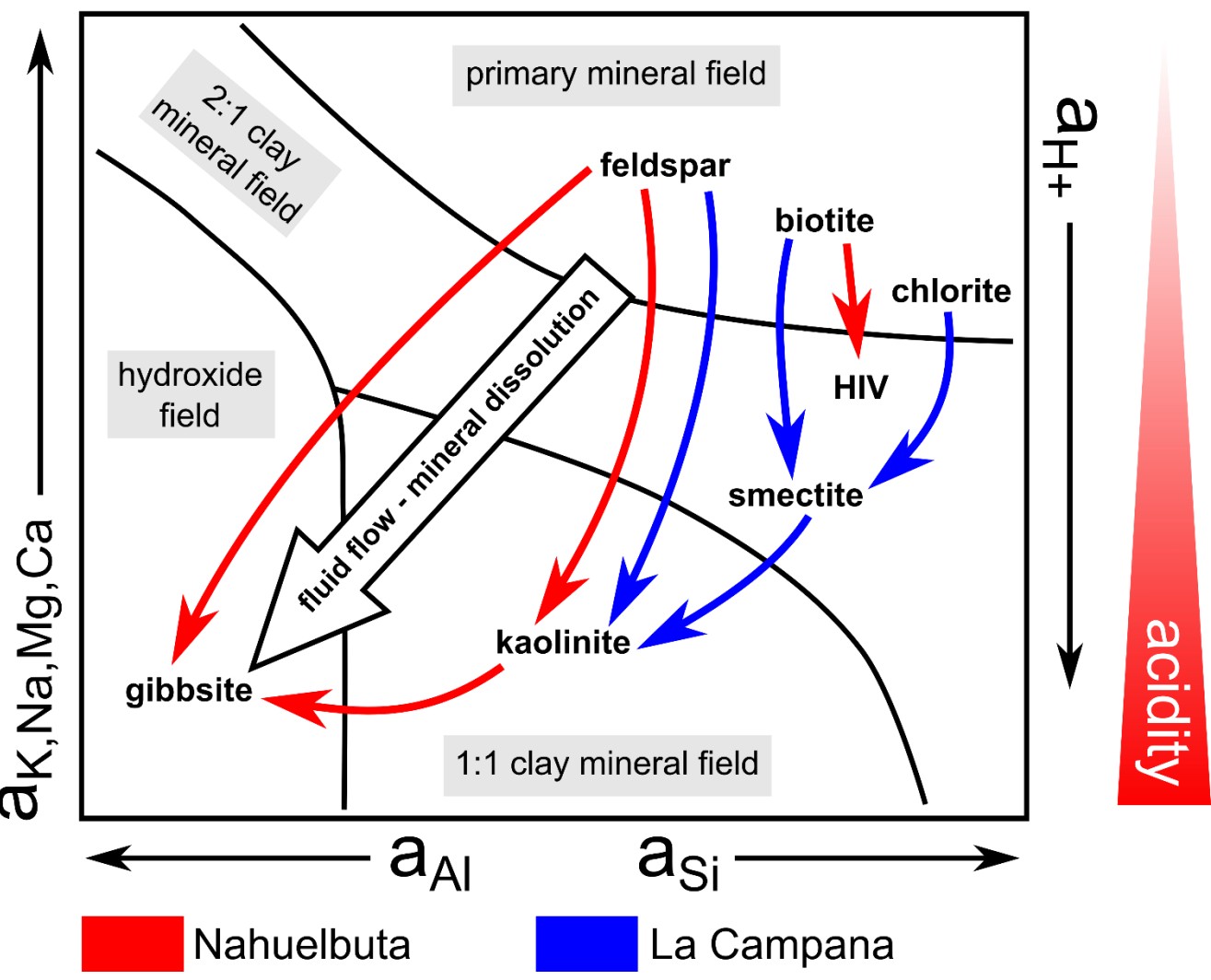

 **Figure 9: Schematic diagram showing the transformation of primary minerals to secondary minerals (clay minerals and aluminium**
**hydroxide) depending on the activities of H+, Si, Al, K, Na, Mg and Ca. The depletion of the alkali and alkaline earth metals, and**
**the increase of the Al activity are coupled to an increase of the mineral dissolution and the removal of solutes by a higher subsurface**
**fluid flow. Elevated $a_{H+}$-values (i.e., lower pH) increase the mineral solubility. Modified from Chesworth et al. (2008). HIV = hydroxy-**
**interlayered vermiculite, a = thermodynamic activity**

## 5.2 Weathering-intensifying processes

### 5.2.1 Porosity increase by weathering-induced fracturing and its impact on the weathering depth

Ferrous primary minerals of the LC granodiorite can frequently be identified as initiating locations of micro-cracks. This observation can be related to weathering-induced fracturing (WIF) due to the increase in volume caused by the oxidation of Fe(II) in Fe(II)-bearing silicates (e.g., Buss et al., 2008; Behrens et al., 2015; Kim et al., 2017) and the formation of secondary Fe(III) oxyhydroxides (Fletcher et al., 2006; Lebedeva et al., 2007; Anovitz et al., 2021; Fig. 3d). This process generates and increases surface areas of primary minerals and in turn accelerates weathering reactions (positive feedback between the formation of secondary minerals and the infiltration of fluids (esp. $O_2$ and water) to depth; e.g., Røyne et al., 2008). These weathering-induced fractures consequently facilitate the presence of surface-derived $O_2$ in the deep subsurface (Kim et al., 2017) and the corresponding transport through the saprolite/soil is dominated by advection (Lebedeva et al., 2007). The bedrock of LC is richer in Fe-bearing minerals than the investigated granite of NA (ca. 25 wt.% in LC and ca. 10 wt.% in NA) and hosts biotite, hornblende, chlorite, and magnetite as Fe(II) sources. A considerable amount of the total Fe content is bound in magnetite (roughly 0.7 wt.% of the total $Fe_2O_3$ content if the magnetite content of the bulk sample is 1 wt.%). However, we found no microscopic evidence (no oxidation) nor indications in the magnetic susceptibility results that the Fe(II) in magnetite is available for weathering reactions. Thus, we conclude that magnetite is stable under the environmental conditions of LC. Of the three remaining Fe(II)-bearing minerals, biotite was found to be the most important one for the generation of WIF in LC (see also Buss et al., 2008; Bazilevskaya et al., 2013, 2015) due to its volumetric expansion during weathering (e.g., Goodfellow et al., 2016). Although WIF also occurs in NA (Fig. 4b) it does not seem to significantly increase the permeability of the rock which can be related to the low Fe(II) content of the dominant bedrock ($Fe_2O_3$ (total Fe): <3 wt.%; Table S1; see Kim et al., 2017).

Other than that, chlorite is suggested to be an important mineral in the development of the investigated weathering profile in LC. The original chlorite content of the bedrock in LC has been completely transformed into interstratified chlorite-smectite in the soil pit profile. We suggest that this transformation plays a significant role for the development of the LC profile since expandable clay minerals are known to disaggregate rock by swelling (e.g., Dunn and Hudec, 1966; Jiménez-González et al., 2008). The ensuing fracturing also forms new fluid pathways and new access to reactive surfaces of primary minerals which in turn fosters weathering reactions (positive feedback mechanism; see e.g., Røyne et al., 2008). Even though expandable clay minerals can also cause sealing of the subsurface (Kim et al., 2017) we do not regard this effect as significant for LC since clay contents are very low. However, a minor retardation of the fluid flow from surface to depth due to the expansion of the

interstratified clay minerals in LC (Fig. 8a) cannot be excluded (see Kim et al., 2017). In conclusion, we propose that small
amounts of expandable clay minerals like in LC can generate porosity whereas high amounts of expandable clay minerals can
reduce porosity.
The feedback mechanism of weathering-induced fracturing is presented here for granodiorite. However, the significance of
this mechanism is not restricted to plutonic rocks. Weathering-induced fracturing requires Fe(II)-bearing minerals such as
biotite and/or potentially the presence of expandable clay minerals that cause the formation of cracks by volume increase
during weathering. This feedback concept is thus transferable to all igneous, metamorphic and sedimentary rocks that contain
these minerals.

### 473  5.2.2 Increase of weathering intensity by biogenic activity

The formation of secondary minerals such as clay minerals and aluminium hydroxide is among other factors controlled by
biogenic activity since organic acids and an acidity-increase by elevated organic-derived $CO_2$ contents accelerate dissolution
rates of primary minerals (see e.g., Lucas, 2001; Lawrence et al., 2014). This effect needs to be considered for the organic-rich
and acidic subsurface of NA (see Bernhard et al., 2018). The acidity likely contributes to the high degree of mineral dissolution
in NA (see $a_{H+}$ in Fig. 9), which consequently leads to an increased formation of secondary minerals.
The depth interval 120–140 cm in LC is characterized by lower amounts of biotite and a different clay mineral composition
compared to the surrounding depth intervals (Fig. 7b; Fig. 8a). This depth interval coincides with a discontinuity crossing the
entire profile width (Fig. 1d). We interpret this plant-root-containing discontinuity in the saprolite as a fracture remnant since
there are no indications of a lithological heterogeneity in this zone (e.g., a significant change in the magnetic susceptibility or
of the primary mineral content; Fig. 7b,c). To explain the lower biotite content and the different clay composition in this part
of the profile, we propose an intensification of weathering reactions in the vicinity of the fracture fostered by the observed
plant roots (e.g., Fimmen et al., 2008; Pawlik et al., 2016; Nascimento et al., 2021). This weathering-promoting mechanism
might account for the increase in interstratified chlorite-smectite and the appearance of interstratified mica-vermiculite (Fig.
8a), while the amount of biotite decreases due to its transformation to secondary minerals (Fig. 7b).

### 488  5.3 Weathering-mitigating processes

### 489  5.3.1 $O_2$ consumption by Fe-bearing silicates and its impact on the weathering depth and intensity

The granodiorite of LC hosts an abundance of Fe(II)-bearing minerals (Fig. 7b). The $Fe_2O_3$ content of the LC bedrock after
subtraction of the inert magnetite-bound $Fe_2O_3$ fraction (since 100 % pure magnetite contains 69 % $Fe_2O_3$, 0.94 % magnetite
as analysed in the LC bedrock equals to 0.65 % magnetite-bound $Fe_2O_3$ which needs to be subtracted) is 5.34 wt.% (for
comparison: 2.33 wt.% $Fe_2O_3$ in NA). Since $O_2$ is reduced by the oxidation of mineral-bound Fe(II) (e.g., White and Yee,
1985; Perez et al., 2005) and the consequent formation of secondary minerals, the $O_2$ content and hence oxidative weathering
reactions are expected to decrease from surface to depth. A rapid decrease of the $O_2$ concentration to depth is characteristic for

weathering systems in which $O_2$ transport is dominated by diffusion (Behrens et al., 2015). Given the observed deep fracturing due to Fe(II) oxidation (i.e., WIF) in LC and the consequent deep connectivity between the surface and the subsurface (Kim et al., 2017), the $O_2$ transport in LC is most likely dominated by advection. As a consequence, diffusive $O_2$ transport is insignificant in the upper regolith of LC and the $O_2$ consumption by Fe(II) oxidation is not limiting the regolith depth in LC (compare Bazilevskaya et al., 2013). It has been argued that WIF and thus a thicker regolith is more likely when the ratio $pO_2/pCO_2$ in soil water is greater than the ratio of the capacity for $O_2$ consumption to the capacity for $CO_2$ consumption in bedrock (Stinchcomb et al., 2018). In the study sites, decomposition of organic matter is restricted to the topsoil, likely because organic matter at depth becomes stabilized against microbial decomposition (Scheibe et al., 2023). Thus, we suggest that the $pCO_2$ of water in the deeper profile part of LC is low (i.e., $pO_2/pCO_2$ is high), and $O_2$ is not being consumed by organic matter decomposition but is available for Fe(II) oxidation and hence WIF. The WIF-controlled connectivity between the surface and the subsurface results in an $O_2$ availability for oxidative weathering processes at great depth. On the other hand, the weak chemical weathering in LC is in good agreement with the low precipitation rate (~350 mm yr$^{-1}$; Übernickel et al., 2020). The low precipitation rate entails a small infiltration of water to depth and hence minor primary mineral dissolution and thus chemical weathering at depth.

The cracks around weathered biotite in the investigated samples of LC are (mainly) filled with Fe(III) oxyhydroxides as revealed by the high Fe-enrichment detected in electron microprobe maps (Fig. 3d). Newly formed weathering-induced fractures make the biotite more accessible to surface inputs like water and $O_2$ which promotes the dissolution of biotite. The solutes formed as a result migrate along the weathering-induced cracks and precipitate in the vicinity of the biotite crystal as secondary phases (Fig. 3c). Thus, we propose that the reactive surface of biotite is partly shielded from weathering reactants (water, $O_2$) due to the precipitation of secondary minerals (see e.g., Navarre-Sitchler et al., 2015; Vázquez et al., 2016; Gerrits et al., 2020; 2021). Combined with the low subsurface water availability in LC causing a low mineral dissolution degree, this shielding might contribute to the relatively stable biotite content throughout the LC profile (Table S4).

**5.3.2 Reduction of weathering intensity and -depth by damping of fluid flow**

The formation of secondary minerals such as clay minerals (via amorphous and poorly crystalline precursors; see Fig. 6) can decrease the porosity (e.g., Bazilevskaya et al., 2015; Navarre-Sitchler et al., 2015) formed by WIF and dissolution. Al-rich phases were found as precipitates in partly dissolved plagioclase of NA (Fig. 4d–f) and within cracks which can often be identified as weathering-induced. We suggest that the abundant presence of clay minerals and gibbsite in NA restricts the fluid flow through such fractures and pores. The clay-rich zone in the uppermost metre of the NA soil pit profile (around 50 cm depth; Fig. 7d) likely acts as a (partially) shielding horizon (impeding vertical flow of surface inputs to the deep subsurface; see e.g., Lohse and Dietrich, 2005). Clay-rich horizons can therefore influence the dynamic of the subsurface fluid flow and thus mitigate mineral transformations and chemical weathering at depth. At the same time, these conditions foster a long fluid residence time in the upper regolith and thus promote the precipitation of secondary minerals such as clay minerals that may impel the weathering of primary minerals in the upper part of the weathering profile (see Maher, 2010). However, the seasonal

sealing of fractures and pore spaces due to an increase of soil moisture and an ensuing clay expansion (Kim et al., 2017) is not
assumed for NA as expandable secondary minerals barely occur in the clay-size fraction of NA (Fig. 8b).
The negative feedback mechanism presented here is demonstrated for granite. However, the concept is essentially based on
newly formed minerals such as clay minerals that inhibit the subsurface fluid flow by blocking pathways. This feedback
mechanism can thus be significant for weathering systems developing from all igneous, metamorphic and sedimentary rocks
where secondary minerals can block the permeable porosity formed by WIF and primary mineral dissolution.

## 6 Conclusions

In two 6 m deep weathering profiles formed on granitic rock in two climatic zones (Mediterranean and humid climate), we
found different degrees of elemental loss by chemical weathering, and different secondary minerals. Under Mediterranean
climate conditions (La Campana), Fe(II)-oxidation, precipitation of Fe(III) oxyhydroxide and clay swelling lead to fracturing
and the formation of fluid pathways. This weathering-induced fracturing (WIF) is likely one of the dominant controls on the
development of the upper regolith as it leads to a deep infiltration of surface inputs (esp. water and $O_2$) which in turn causes
further WIF. While the intensity of chemical weathering at the Mediterranean site is low, it was detected throughout the entire
6 m deep profile. This suggests that the weathering front is located at much greater depth in La Campana. The overall low
abundance of secondary minerals can be explained by the low climate-related subsurface water availability in La Campana.
The lack of large quantities of secondary minerals ensures that fractures and porosity generated by WIF remain accessible to
water and gases. Thus, we conclude that the development of the deep but weak chemical weathering in the upper regolith of
La Campana is significantly controlled by two mechanisms: (1) A positive feedback loop between the formation of secondary
minerals and the infiltration of fluids to depth induced by (mainly) biotite weathering (WIF) which leads to a deep surface-
subsurface connectivity for weathering reactants (in particular $O_2$). (2) Low subsurface water availability resulting in a low
amount of secondary minerals which would otherwise seal this connectivity.
Under humid climate conditions (Nahuelbuta), clay minerals, gibbsite as well as amorphous and poorly crystalline secondary
minerals largely formed due to intense plagioclase dissolution. We link this intense dissolution to the high climate-related
subsurface water availability in Nahuelbuta. The secondary minerals thus formed are suggested to impede the flow of surface
inputs to depth. Moreover, the generally lower amount of Fe(II)-bearing silicates in Nahuelbuta compared to La Campana
results in less WIF and thus fewer fluid pathways. Therefore, we conclude that the development of the weathering profile in
Nahuelbuta is predominantly governed by two mechanisms: (1) Considerable climate-related subsurface water availability and
high biogenic activity which lead to intense weathering of primary minerals in the upper part of the regolith. (2) A negative
feedback loop between the formation of secondary minerals and the infiltration of fluids to depth induced by (mainly)
plagioclase weathering and the ensuing formation of secondary minerals which leads to a poor surface-subsurface connectivity
for weathering reactants. The main findings and factors that are most relevant to the development of the different weathering
systems are summarized in Figure 10.
The relationship between precipitation and the degree of chemical weathering along the climate gradient of the Chilean Coastal
Cordillera was found to be non-linear and non-systematic (Oeser and von Blanckenburg, 2020; Schaller and Ehlers, 2022).
We argue that a systematic relationship is likely concealed by variations in the mineral content of the bedrocks and the
associated feedback mechanisms. However, the investigated feedbacks provide a causal explanation for the depth of chemical
weathering. This study illustrates how the formation of secondary minerals and the infiltration of surface-derived fluids to
depth are interlinked by positive and negative feedback loops. We demonstrated that these feedback loops and the climatic
conditions they occur in are important controls on the development of the upper regolith.

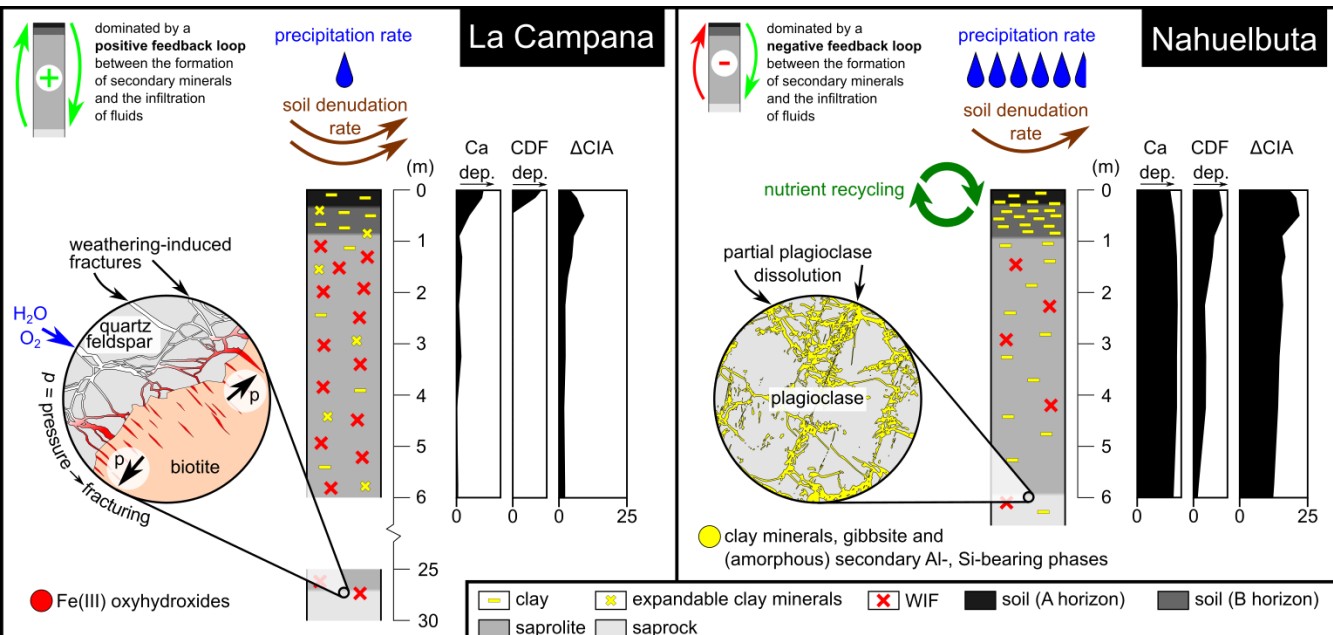


**Figure 10: Schematic summary of the two weathering systems. According to our findings, the regolith of La Campana (LC) is**
**dominated by a positive feedback loop between weathering-induced fracturing (WIF) and the infiltration of fluids to depth. WIF**
**creates deep-reaching pathways for fluids (water, O₂) and hence a good connectivity between the surface and the subsurface.**
**Moreover, the low water availability in the Mediterranean climate inhibits the formation of large amounts of secondary minerals**
**(i.e., low weathering intensity) that could seal these pathways. The high denudation rate in LC results in a short residence time of**
**weathered material in the profile and could therefore contribute to the detected lower weathering intensity (i.e., less chemical**
**weathering). The regolith of Nahuelbuta (NA), on the other hand, was found to be dominated by a negative feedback loop between**
**the formation of secondary minerals and amorphous phases, and the infiltration of fluids to depth. These secondary solids are**
**consequences of the high water availability in NA that results in intense chemical weathering (i.e., high weathering intensity). The**
**high weathering intensity entails the formation of abundant secondary minerals and amorphous phases that reduce the connectivity**
**between the surface and the subsurface. The lower denudation rate and thus longer residence time of weathered material in NA**
**likely contributes to the more intense chemical weathering. dep. = depletion.**

## Data availability

Datasets related to this article can be found in the data publication Hampl et al. (2022b). The data publication is hosted at the GFZ Data Services and can be downloaded by clicking on "Download data and description" in the field "Files".

## Sample availability

The IGSN-registered samples used in this article are deposited at the Department of Applied Geochemistry (Technische Universität Berlin) and are listed in the data publication of this paper (Hampl et al., 2022b).

## Author contribution

Ferdinand J. Hampl: conceptualization, methodology, investigation, writing – original draft preparation

Ferry Schiperski: methodology, supervision, writing – review & editing

Christopher Schwerdhelm: investigation, writing – review & editing

Nicole Stroncik: investigation, writing – review & editing

Casey Bryce: funding acquisition, writing – review & editing

Friedhelm von Blanckenburg: supervision, writing – review & editing

Thomas Neumann: funding acquisition, supervision, writing – review & editing

## Competing interests

The authors declare that they have no conflict of interest.

## Acknowledgements

This work was supported by the German Research Foundation (DFG) priority research program SPP-1803 "EarthShape: Earth Surface Shaping by Biota" (grant number NE 687/9-1) and the EarthShape Coordination (EH 329/17-2, BL562/20-1). We are grateful to Dr. Kirstin Übernickel for the management of the drilling campaigns and to Prof. Andreas Kappler for his support. We would also like to thank Michael Facklam for his help in determining the clay content and Dr. Katja Emmerich for her valuable hints on the clay mineralogy. The authors would also like to thank Prof. Dr. Peter Finke, Prof. Dr. Veerle Vanacker

and Prof. Dr. Susan L. Brantley for their valuable comments and suggestions that greatly improved the manuscript. Finally,
we are grateful to Antonia Roesrath for her help in registering the samples.

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
