# Peer review of "Feedbacks between the formation of secondary minerals and the infiltration of fluids into the regolith of granitic rocks in different climatic zones (Chilean Coastal Cordillera)"

_Earth Surface Dynamics, 2022_

## Author Response (AR1)

**Response to Prof. Dr. Peter Finke**

Thank you for your constructive feedback and annotations on our manuscript. In our point-by-point response your comments are in italics and our responses are in bold. The line numbers refer to the original preprint version you have commented.

*This well-written manuscript is an important extension to existing insights because it focuses on the depth and intensity of weathering, and not only the strength of weathering, as a function of climate.*

**Thank you for your appreciation of the manuscript.**

*(i) in terms of the processes, the production rate of swelling clay minerals and the fracturing rate by weathering of some Fe-bearing minerals, triggered by climate, lead in the hypothesis behind this research to either positive or negative feedbacks. The processes themselves are not different between the 2 extremes sketched (i.e. fracturing dominates or pore blocking dominates), rather the process rates, and intermediate situations may also exist, e.g. close to the equivalence point. Schaller&Ehlers (2022) inventoried whole-profile (often quite shallow) CDF's for different annual precipitation amounts, a.o. for similar sites in Chile. García-Gamero et al, 2022, fig.10, linked these whole-profile CDF's back to the Albrecht (1957) curve and found a pattern of response to precipitation which could in the current manuscript be discussed, linking precipitation to the underlying processes. As in the current manuscript precipitation is either 346 (LC) or 1927 (NA) mm.y-1, only fairly extreme points in this continuum are sampled and I wonder how the authors envisage intermediate situations in terms of precipitation (e.g. at 800 mm).*

**Your question about settings characterized by an intermediate precipitation rate is very interesting and we are grateful for the publications you recommended.**
**We hypothesize that weathering of Fe(II)-bearing minerals such as biotite and swelling of expandable clay minerals (e.g., interstratified chlorite-smectite) lead to a positive feedback (fracturing → more fluid infiltration) like in LC, while non-expandable clay minerals (e.g., kaolinite) and amorphous phases lead to a negative feedback (blocking of fluid pathways → less fluid infiltration) like in NA. These feedbacks may contribute to the non-linear and non-systematic dependence of chemical weathering on precipitation that has been found in the Chilean Coastal Cordillera (Oeser and von Blanckenburg, 2020; Schaller and Ehlers, 2022). Moreover, minor variations in the primary mineral**

content of the bedrock (e.g., amount of Fe(II)-bearing minerals) can have a substantial impact on the development of the regolith. We found mineralogical differences between the bedrocks of LC and NA that affect the development of the respective regoliths (clarified in our discussion section). Thus, we assume that a systematic correlation between chemical weathering and precipitation on a global scale is heavily confounded by differences in the mineral content of the bedrocks considered.

The core finding presented in Albrecht (1957) and García-Gamero et al. (2022), is a parabola-like function between precipitation and chemical weathering. However, a non-linear/non-systematic relation between chemical weathering and mean annual precipitation was found in the Chilean Coastal Cordillera and has been explained by vegetation (Oeser and von Blanckenburg, 2020; Schaller and Ehlers, 2022). Based on these previous studies, we do not believe that an intermediate case – with a precipitation rate between LC and NA – would be characterized by more intense weathering as suggested in Albrecht (1957) and García-Gamero et al. (2022). Therefore, we also do not believe that the conditions in an intermediate case can be interpolated from our results in LC and NA.

However, if this aspect is considered to be beneficial to the manuscript, we suggest adding the following text to the conclusion (starting in line 521):

The relationship between precipitation and the degree of chemical weathering along the climate gradient of the Chilean Coastal Cordillera was found to be non-linear and non-systematic (Oeser and von Blanckenburg, 2020; Schaller and Ehlers, 2022). We argue that a systematic relationship is likely concealed by variations in the mineral content of the bedrocks and the associated feedback mechanisms. However, the investigated feedbacks provide a causal explanation for the depth of chemical weathering.

*(ii) In NA and LC exposure to oxygen is different because the fracturing in LC leads to "open" fractures while in NA these are less "open". This leads to higher oxydative stress in LC, producing FeIII-minerals. Fracturing could on the other hand also lead to higher contact area and residence time of infiltrating water in NA, which will increase weathering as appears from CDF and delta CIA. This could be added to the discussion.*

Thank you for the comment. We agree with your suggestion and will add the following sentence to our text (starting in line 494):

At the same time, these conditions foster a long fluid residence time in the upper regolith and thus promote the precipitation of secondary minerals such as clay minerals

**that may impel the weathering of primary minerals in the upper part of the weathering profile (see Maher, 2010).**

*(iii) The type of clay minerals, either newly formed or transformed, leads to swelling behavior (CF) or pore-blocking behavior (NA). What is the linkage to the (Si-rich) parent materials studied here (which are somewhat different, c.f. lines 373-376). Also the clay content (higher in NA) is stated to have a clear influence. I would appreciate some discussion on (a) to what degree are the mechanisms portabe to other rock types? and (b) are the differences in clay content a function of the degree of weathering or of the parent materials?*

    **(a) This aspect was clearly missing in our manuscript – thank you for pointing that out. Our study focusses on granitic rocks, but we see the necessity to consider the investigated mechanisms for other rock types as well. Therefore, we will add the following to the discussion:**

**Starting in line 445: The feedback mechanism of weathering-induced fracturing is presented here for granodiorite. However, the significance of this mechanism is not restricted to plutonic rocks. Weathering-induced fracturing requires Fe(II)-bearing minerals such as biotite and/or potentially the presence of expandable clay minerals that cause the formation of cracks by volume increase during weathering. This feedback concept is thus transferable to all igneous, metamorphic and sedimentary rocks that contain these minerals.**

**Starting in line 496: The negative feedback mechanism presented here is demonstrated for granite. However, the concept is essentially based on newly formed minerals such as clay minerals that inhibit the subsurface fluid flow by blocking pathways. This feedback mechanism can thus be significant for weathering systems developing from all igneous, metamorphic and sedimentary rocks where secondary minerals can block the permeable porosity formed by WIF and primary mineral dissolution.**

    **(b) That is a crucial question and we also think that it deserves more attention. We will add the following to our text (starting in line 407):**

        **The mineral composition of the clay-size fraction (Fig. 8) is dependent on the bedrock composition (e.g., more chlorite in LC) and the climate-dependent mineral dissolution (see Fig. 9) in the study sites. However, we argue that the**

**amount of secondary minerals is largely a function of the climatic conditions that control the weathering intensity via water availability in the study sites.**

*l.383: in NA nutrient recycling is more important than uptake of nutrients released by weathering: It could be specified that the higher precipitation will likely lead to higher biomass production by plants in NA, which will produce more litter, stimulate its biogenic decay and thus enhance nutrient cycling in the biologically active zone. A similar mechanism is already mentioned in 5.2.2 in the context of production of acidity via CO2-release.*

**As this is such an important piece of information in the weathering system of Nahuelbuta, we agree with your advice of emphasizing it more. Thus, we reformulated your statement and we will add the following to our text (starting in line 381):**

**We assume that the high precipitation rate in NA leads to more biomass production by plants, which in turn implies more litter production and a stimulation of biogenic decay that supplies plants with nutrients.**

*l.387: Al(OH) > Al(OH)3*

**Thank you for pointing that out. We corrected this mistake in line 378.**

Thanks again for your review and we hope that our additions do justice to your suggestions for improvement.

**References**

Albrecht, W. A.: Soil Fertility and Biotic Geography, Geogr. Rev., 47, 86, 47, 86–105, https://doi.org/10.2307/212191, 1957.

García-Gamero, V., Vanwalleghem, T., Peña, A., Román-Sánchez, A., and Finke, P. A.: Modelling the effect of catena position and hydrology on soil chemical weathering, SOIL, 8, 319–335, https://doi.org/10.5194/soil-8-319-2022, 2022.

Maher, K.: The dependence of chemical weathering rates on fluid residence time, EPSL, 294, 1–2, 101-110, https://doi.org/10.1016/j.epsl.2010.03.010, 2010.

Oeser, R. A., and von Blanckenburg, F.: Do degree and rate of silicate weathering depend on plant productivity? Biogeosciences, 17, 4883‑4917, https://doi.org/10.5194/bg-17-4883-2020, 2020.

Schaller, M., and Ehlers, T. A.: Comparison of soil production, chemical weathering, and physical erosion rates along a climate and ecological gradient (Chile) to global observations, Earth Surf. Dynam., 10, 131–150, https://doi.org/10.5194/esurf-10-131-2022, 2022.

**Response to Prof. Dr. Susan Brantley**

Thank you for your constructive feedback and comments on our manuscript. In our point-by-point response your comments are in italics and our responses are in bold. The line numbers refer to the original preprint version you have commented.

*This is a very fine, high-quality discussion of two weathering profiles. The main point is to try to argue that clays and weathering induced fracturing create feedbacks that affect depth of weathering. The figures and diagrams are absolutely, superbly beautiful and the paper is well written in general. The paper should be published after minor revision. I am very interested in this topic and I suggest a few papers we wrote that might be helpful: if they are not helpful, no need to cite them. Whatever helps to explain the data should be cited.*

**Thank you for your appreciation of the manuscript and your motivating words.**

*My most important point is that the authors should consider making a final schematic or diagram that summarizes the two profiles and explains the differences and why one profile is deeper than the other. This figure should summarize all the main points and whatever the authors deem as most important: the authors could summarize relative rainfall, relative erosion rate, relative extent of WIF, relative extent of depletion, depth of total weathering, etc., and indicate which variables explain the difference in depths or extents of weathering . This will help the reader carry away the main message.*

**Thank you for the suggestion. The new conclusion cartoon summarizes the main points and factors contributing to the situation in the two study sites (see attachment "Figure 10"). We will add the following sentence to introduce the new Figure 10 (Line 521):**

**"The main findings and factors that are most relevant to the development of the different weathering systems are summarized in Figure 10."**

[Figure]

**Figure 10: Schematic summary of the two weathering systems. According to our findings, the regolith of La Campana (LC) is dominated by a positive feedback loop between weathering-induced fracturing (WIF) and the infiltration of fluids to depth. WIF creates deep-reaching pathways for fluids (water, O₂) and hence a good connectivity between the surface and the subsurface. Moreover, the low water availability in the Mediterranean climate inhibits the formation of large amounts of secondary minerals (i.e., low weathering intensity) that could seal these pathways. The high denudation rate in LC results in a short residence time of weathered material in the profile and could therefore contribute to the detected lower weathering intensity (i.e., less chemical weathering). The regolith of Nahuelbuta (NA), on the other hand, was found to be dominated by a negative feedback loop between the formation of secondary minerals and amorphous phases, and the infiltration of fluids to depth. These secondary solids are consequences of the high water availability in NA that results in intense chemical weathering (i.e., high weathering intensity). The high weathering intensity entails the formation of abundant secondary minerals and amorphous phases that reduce the connectivity between the surface and the subsurface. The lower denudation rate and thus longer residence time of weathered material in NA likely contributes to the more intense chemical weathering. dep. = depletion.**

*We have argued that the depth of a steady state weathering profile is determined by the depth when the weathering advance rate has decreased and it finally equals the erosion rate (Lebedeva and Brantley, 2020). This reflects the idea that the vertical advection rate of meteoric water is higher at the surface than at depth because at various depths where the permeability contrast is high, water moves laterally out of the profile (Rempe and Dietrich, 2014; Braun et al., 2016; Lebedeva and Brantley, 2018; Harman and Kim, 2019; Reis and Brantley, 2019). Since the weathering advance rate is a function of the vertical advective velocity, the weathering advance rate decreases downward as more and more water is lost to lateral flow. The idea is that the depth of a profile reflects the amount of time it takes for the*

*weathering profile to reach that steady state: as a profile deepens with time (before steady state), more and more water is lost to lateral flow and the vertical weath advance rate slows until it equals the erosion rate (Lebedeva and Brantley, 2020). Is it possible that whichever profile is deeper, it just took longer to deepen to the point that the weathering rate at the bottom of the profile finally slowed to the local erosion rate? If your observations contradict our hypothesis, by all means point that out. And if this conceptual model does not help, no need to cite any of these papers.*

*If you do not like that explanation, perhaps this model (Reis and Brantley, 2019) might be helpful? In those simulations, some of the profiles are very low in weathering extent but very deep, somewhat like the LC profile. I have never thoroughly understood why this happens.*

**This is an interesting hypothesis. However, for two reasons we are not able to fully evaluate this concept. Firstly, we do not know whether our profiles reached steady state or if they are in a transitional state. Secondly, we do not consider the whole weathering profile in this manuscript. Thus, we cannot comment on the interplay between weathering advance rate and erosion rate. However, your comment inspired us to evaluate the interplay between weathering intensity and denudation rate – a topic that was not addressed sufficiently in our text. As the denudation rate in LC is generally higher than in NA, we assume that the residence time of weathered material in the profile of LC is shorter than in NA. Thus, there is less time for chemical weathering in LC. In combination with the lower water availability in LC, this might further contribute to the lower weathering intensity in the regolith of LC compared to NA. We will introduce this likely effect of different denudation rates by adding the following to our text (Line 408):**

**"As this study does not consider the entire weathering profile in LC and NA, the interplay between erosion rate and weathering advance rate (Lebedeva and Brantley, 2020) is not addressed here. However, the different denudation rates in the study areas (mean soil denudation rate in LC: ~ 61, in NA: ~ 33 t km$^{-2}$ yr$^{-1}$; Oeser et al., 2018) likely affect the weathering intensity. Due to the higher denudation rates in LC compared to NA (Oeser et al., 2018; van Dongen et al., 2019), we hypothesize that the residence time of weathered material in the regolith of LC is shorter than in NA. Thus, there is less time for chemical weathering in LC. In combination with the lower water availability in LC, this factor might contribute to the lower weathering intensity in the regolith of LC compared to NA. NA, on the other hand, is characterized by a longer residence time of weathered material. Together with the higher water availability, this factor might contribute to the high weathering intensity in the upper regolith of NA. The situation in**

**LC may be comparable to an incompletely developed profile, and the situation in NA to a completely developed profile (Reis and Brantley, 2019)."**

*The LC profile is an incompletely developed profile with respect to many of the elements...this is often interpreted to mean that erosion rate is comparatively high or the profile has not reached steady state. However, smectite can retain base metal cations at the land surface even at steady state (where weathering advance rate = erosion rate) (Brantley et al., 2023) so that a profile may not reach 100% depletion wrt cations. Perhaps the authors want to point this out…that one of the big differences in the tau plots is that LC is incompletely developed while NA is completely developed wrt base metal cations, but this may be best explained simply because smectite formation is causing cation retention at the surface, pinning the tau plot so that it will not go to 100% depletion.*

**This is a very interesting concept. However, based on the relatively low content of smectitic clay minerals in LC, we do not think that they are the major reason why depletion does not or cannot reach 100%. We attribute the low depletion mainly to the low water availability in the subsurface of LC. Moreover, due to the low weathering intensity in LC, base-metal-cation-containing primary minerals are far more abundant than smectitic clay minerals which makes it difficult to estimate their influence on the tau plots.**

*Di the authors explain why Zr is the best immobile element?*

**Our decision to use Zr instead of Nb is justified in the data publication:**
**"Zr (e.g., Harden, 1987), Ti, and Nb (e.g., Brimhall and Dietrich, 1987) are regarded as immobile (i.e., conservative) elements during weathering and are therefore used for calculating weathering indices. The usage of Nb was rejected here since the sample contents which were analysed with powder pellets do not exceed 10 ppm (see Table S1) and the sensitivity of the method used is generally not sufficient to measure low element contents. Zr was used here as an immobile element for the calculation of the depletion by weathering."**
**We did not use Ti as an immobile element because Ti can be structurally included in biotite, which is oxidized and dissolved during weathering. Unfortunately, this explanation is missing in our text. We will thus add the following sentence to the data publication:**

**"Ti was discarded as an immobile element for the calculations, as Ti contents are common in biotite which is oxidized and dissolved in the profiles."**

*I don't think that Figure 9 is a phase diagram. I think it is an activity-activity diagram.*

**We will delete "phase" in the first sentence of the figure caption.**

*I have hypothesized that WIF occurs when oxidation occurs deeper than dissolution (Stinchcomb et al., 2018). If this is true, then the protolith's capacity to reduce O2 [for example, the Fe(II) content] and O2 availability should affect WIF. Stinchcomb et al. argued that if the ratio of pO2 to pCO2, in soil water, R'(aq), is greater than the ratio of the capacity of the protolith to consume O2 and CO2, R0, then WIF is more likely to occur. We showed this to be helpful in interpreting several weathering profiles. In that paper we also observed the presence of smectite in some of our granite/diabase profiles and not others. I think the authors should at least take a look at Stinchcomb et al., and perhaps consider whether any of the ideas in that paper (which got pretty complex unfortunately) are helpful or not, or whether the profiles are different or similar to the ones studied here.*

**The concept in Stinchcomb et al. (2018) is a great addition to our interpretation of the weathering system in LC. In combination with recently published data on our profiles, we will add the following sentence to our manuscript text (Line 473):**

**"It has been argued that WIF and thus a thicker regolith is more likely when the ratio $pO_2/pCO_2$ in soil water is greater than the ratio of the capacity for $O_2$ consumption to the capacity for $CO_2$ consumption in bedrock (Stinchcomb et al., 2018). In the study sites, decomposition of organic matter is restricted to the topsoil, likely because organic matter at depth becomes stabilized against microbial decomposition (Scheibe et al., 2023). Thus, we suggest that the $pCO_2$ of water in the deeper profile part of LC is low (i.e., $pO_2/pCO_2$ is high), and $O_2$ is not being consumed by organic matter decomposition but is available for Fe(II) oxidation and hence WIF."**

**The discussion on the effect of smectite in Stinchcomb et al. (2018) is based on Kim et al. (2017) which we cite on several occasions in our manuscript.**

Thanks again for your review and we hope that our answers do justice to your suggestions for improvement.

**References**

Brimhall, G. H. and Dietrich, W. E.: Constitutive mass balance relations between chemical composition, volume, density, porosity, and strain in metasomatic hydrochemical systems: Results on weathering and pedogenesis. Geochim. Cosmochim. Acta, 51, 567-587, https://doi.org/10.1016/0016-7037(87)90070-6, 1987.

Harden, J. W.: Soils developed in granitic alluvium near Merced, California. U.S. Geol. Surv. Bull. 1590-A, https://doi.org/10.3133/b1590A, 1987.

Kim, H., Stinchcomb, G., and Brantley, S.: Feedbacks among $O_2$ and $CO_2$ in deep soil gas, oxidation of ferrous minerals, and fractures: A hypothesis for steady-state regolith thickness, Earth Planet. Sc. Lett., 460, 29-40, https://doi.org/10.1016/j.epsl.2016.12.003, 2017.

Lebedeva, M. I. and Brantley, S. L.: Exploring an 'ideal hill': how lithology and transport mechanisms affect the possibility of a steady state during weathering and erosion. ESPL, 45(3), 652-665, https://doi.org/10.1002/esp.4762, 2020.

Oeser, R. A., Stroncik, N, Moskwa, L.-M., Bernhard, N., Schaller, M., Canessa, R., van den Brink, L., Köster, M., Brucker, E., Stock, S., Fuentes, J. P., Godoy, R., Matus, F. J., Oses Pedraza, R., Osses McIntyre, P., Paulino, L., Seguel, O., Bader, M. Y., Boy, J., Dippold, M. A., Ehlers, T. A., Kühn, P., Kuzyakov, Y., Leinweber, P., Scholten, T., Spielvogel, S., Spohn, M., Übernickel, K., Tielbörger, K., Wagner, D., and von Blanckenburg, F.: Chemistry and microbiology of the Critical Zone along a steep climate and vegetation gradient in the Chilean Coastal Cordillera, CATENA, 170, 183-203, https://doi.org/10.1016/j.catena.2018.06.002, 2018.

Reis, F. D. A. A. and Brantley, S. L.: The impact of depth-dependent water content on steady state weathering and eroding systems. Geochim. Cosmochim. Acta, 244, 40-55, https://doi.org/10.1016/j.gca.2018.09.028, 2019.

Scheibe, A., Sierra, C. A., Spohn, M.: Recently fixed carbon fuels microbial activity several meters below the soil surface. Biogeosciences, 20, 827-838, https://doi.org/10.5194/bg-20-827-2023, 2023.

Stinchcomb, G. E., Kim, H., Hasenmueller, E. A., Sullivan, P. L., Sak, P. B., and Brantley, S. L.: Relating soil gas to weathering using rock and regolith geochemistry. Am. J. Sci., 318, 727-763, https://doi.org/10.2475/07.2018.01, 2018.

van Dongen, R., Scherler, D., Wittmann, H., and von Blanckenburg, F.: Cosmogenic 10Be in river sediment: where grain size matters and why, Earth Surf. Dynam., 7, 393-410, https://doi.org/10.5194/esurf-7-393-2019, 2019.

**Additions and changes to the manuscript**

This is a structured compilation of the additions (in blue) and changes to the manuscript. The line numbers we give here refer to the revised version (with accepted changes) of our manuscript.

**Line 88-90:** The annual precipitation rate (measured from April 2016 to April 2020) is 346 mm yr$^{-1}$ (Übernickel et al., 2020) and the Holocene net primary production is 280 ± 50 g C m$^{-2}$ yr$^{-1}$ (Werner et al., 2018; Oeser and von Blanckenburg, 2020).

**Line 104-106:** The precipitation rate (measured from end of March 2016 to April 2020) is 1927 mm yr$^{-1}$ (Übernickel et al., 2020) and the Holocene net primary production is 520 ± 130 g C m$^{-2}$ yr$^{-1}$ (Werner et al., 2018; Oeser and von Blanckenburg, 2020).

**Line 378**: (Al(OH)$_3$ = gibbsite).

**Line 381-383:** We assume that the high precipitation rate in NA leads to more biomass production by plants, which in turn implies more litter production and a stimulation of biogenic decay that supplies plants with nutrients.

**Line 409-412:** The mineral composition of the clay-size fraction (Fig. 8) is dependent on the bedrock composition (e.g., more chlorite in LC) and the climate-dependent mineral dissolution (see Fig. 9) in the study sites. However, we argue that the amount of secondary minerals is largely a function of the climatic conditions that control the weathering intensity via water availability in the study sites.

**Line 413-422:** As this study does not consider the entire weathering profile in LC and NA, the interplay between erosion rate and weathering advance rate (Lebedeva and Brantley, 2020) is not addressed here. However, the different denudation rates in the study areas (mean soil denudation rate in LC: ~ 61, in NA: ~ 33 t km$^{-2}$ yr$^{-1}$; Oeser et al., 2018) likely affect the weathering intensity. Due to the higher denudation rates in LC compared to NA (Oeser et al., 2018; van Dongen et al., 2019), we hypothesize that the residence time of weathered material in the regolith of LC is shorter than in NA. Thus, there is less time for chemical weathering in LC. In combination with the lower water availability in LC, this factor might contribute to the lower weathering intensity in the regolith of LC compared to NA. NA, on the other hand, is characterized by a longer residence time of weathered material. Together with the higher water availability, this factor might contribute to the high weathering intensity in the upper regolith of

NA. The situation in LC may be comparable to an incompletely developed profile, and the situation in NA to a completely developed profile (Reis and Brantley, 2019).

**Line 424-425:** Figure 9: Schematic  diagram showing the transformation of primary minerals to secondary minerals (clay minerals and aluminium hydroxide) depending on the activities of H+, Si, Al, K, Na, Mg and Ca.

**Line 461-465:** The feedback mechanism of weathering-induced fracturing is presented here for granodiorite. However, the significance of this mechanism is not restricted to plutonic rocks. Weathering-induced fracturing requires Fe(II)-bearing minerals such as biotite and/or potentially the presence of expandable clay minerals that cause the formation of cracks by volume increase during weathering. This feedback concept is thus transferable to all igneous, metamorphic and sedimentary rocks that contain these minerals.

**Line 493-498:** It has been argued that WIF and thus a thicker regolith is more likely when the ratio $pO_2/pCO_2$ in soil water is greater than the ratio of the capacity for $O_2$ consumption to the capacity for $CO_2$ consumption in bedrock (Stinchcomb et al., 2018). In the study sites, decomposition of organic matter is restricted to the topsoil, likely because organic matter at depth becomes stabilized against microbial decomposition (Scheibe et al., 2023). Thus, we suggest that the $pCO_2$ of water in the deeper profile part of LC is low (i.e., $pO_2/pCO_2$ is high), and $O_2$ is not being consumed by organic matter decomposition but is available for Fe(II) oxidation and hence WIF.

**Line 519-521:** At the same time, these conditions foster a long fluid residence time in the upper regolith and thus promote the precipitation of secondary minerals such as clay minerals that may impel the weathering of primary minerals in the upper part of the weathering profile (see Maher, 2010).

**Line 524-527:** The negative feedback mechanism presented here is demonstrated for granite. However, the concept is essentially based on newly formed minerals such as clay minerals that inhibit the subsurface fluid flow by blocking pathways. This feedback mechanism can thus be significant for weathering systems developing from all igneous, metamorphic and sedimentary rocks where secondary minerals can block the permeable porosity formed by WIF and primary mineral dissolution.

**Line 552-553:** The main findings and factors that are most relevant to the development of the different weathering systems are summarized in Figure 10.

**Line 554-558:** The relationship between precipitation and the degree of chemical weathering along the climate gradient of the Chilean Coastal Cordillera was found to be non-linear and non-systematic (Oeser and von Blanckenburg, 2020; Schaller and Ehlers, 2022). We argue that a systematic relationship is likely concealed by variations in the mineral content of the bedrocks and the associated feedback mechanisms. However, the investigated feedbacks provide a causal explanation for the depth of chemical weathering.

**Line 562-574:**

[Figure]

Figure 10: Schematic summary of the two weathering systems. According to our findings, the regolith of La Campana (LC) is dominated by a positive feedback loop between weathering-induced fracturing (WIF) and the infiltration of fluids to depth. WIF creates deep-reaching pathways for fluids (water, O₂) and hence a good connectivity between the surface and the subsurface. Moreover, the low water availability in the Mediterranean climate inhibits the formation of large amounts of secondary minerals (i.e., low weathering intensity) that could seal these pathways. The high denudation rate in LC results in a short residence time of weathered material in the profile and could therefore contribute to the detected lower weathering intensity (i.e., less chemical weathering). The regolith of Nahuelbuta (NA), on the other hand, was found to be dominated by a negative feedback loop between the formation of secondary minerals and amorphous phases, and the infiltration of fluids to depth. These secondary solids are consequences of the high water availability in NA that results in intense chemical weathering (i.e., high weathering intensity). The high weathering intensity entails the formation of abundant secondary minerals and amorphous phases that reduce the connectivity between the surface and the subsurface. The lower denudation rate and thus longer residence time of weathered material in NA likely contributes to the more intense chemical weathering. dep. = depletion.

**Line 601-602:** The authors would also like to thank Prof. Dr. Peter Finke and Prof. Dr. Susan L. Brantley for their valuable comments and suggestions that greatly improved the manuscript.

---

## Author Response (AR2)

**Response to Prof. Dr. Veerle Vanacker**

Thank you for your constructive feedback and comments on our revised manuscript. In our point-by-point response your comments are in italics and our responses are in bold. The line numbers refer to the manuscript you have commented.

*The revised manuscript adequately addresses the main comments raised by previous reviewers. The manuscript is well written, and nicely illustrated.*

**Thank you for your appreciation of the manuscript.**

*A few minor issues might deserve the authors' attention before publication. They mainly concern the (hydro)climatic characterization of the area and the hydrological control on the weathering reactions.*

*1-Given the focus on the climatic control on chemical weathering, it would be helpful to have more information on the climate regimes in both sites. The soil water balance - and soil water fluxes - is controlled by precipitation, infiltration and evapotranspiration, and temperature and precipitation variability directly affect variation in soil water content. Information on actual evapotranspiration and infiltration is often difficult to collect, but data on the (intra-annual variability of the) precipitation and temperature regimes (and eventually ETp or ETa) would already be helpful to understand the differences in overall water balance between the two sites.*

**Unfortunately, the infiltration and evapotranspiration are unknown for the study sites. However, there is a published set of meteorological station data from the study sites that was collected over a period of several years. We cite this data set in the description of the study sites (Übernickel et al., 2020) but we see the necessity to emphasize it more. Thus, we added the following sentence (Line 90 and 106) to give the reader more information on the climate regimes in both sites:**
**"Records of long-term meteorological data (e.g., precipitation at ground level, soil water content, air temperature, relative humidity) from a weather station near the study site can be found in Übernickel et al. (2020)."**
**We also cite recently published information on MAP and MAT published in Scheibe et al. (2023) (see further below).**

*2-Water availability is cited as an important (hydro)climatic control on weathering, and the authors make a direct link between precipitation, water availability and infiltration/percolation rates (L411-413; L420-422; L501). A simple proxy of soil water balance (like P/ETp) can be helpful to characterize the overall soil water balance & link it with soil water fluxes in the two sites (e.g. Schoonejans et al., 2016). The schematic summary presented in Fig10 illustrates the soil system when ETp and P are roughly in balance. Would it still hold when there is a significant water deficit or surplus (see e.g. discussion in Reis et al. 2017)? In how far differences in soil water balance can have an effect on soil infiltration/percolation and water flow within soil and regolith?*

**Unfortunately, the potential evapotranspiration (ETp) is unknown for the study sites and the necessary parameters to evaluate the soil water balance have not been measured in La Campana and Nahuelbuta. Thus, we cannot provide information on the soil water balance. Our information on water availability and infiltration are solely relative between the study sites. We hope this is acceptable.**

**However, we think that Schoonejans et al. (2016) is a valuable addition to our explanation of the conditions in La Campana. Therefore, we added the following sentence to the manuscript (Line 419):**

**"This would also concur with the finding that water availability in the soil and soil residence time are the limiting factors for weathering processes in dry environments (Schoonejans et al., 2016)."**

*Detailed comments.*

*L31/33: I find it more informative to have the main characteristics (mean annual precip, mean annual temp, mean annual ETp) of both climate regimes ("Mediterranean" and "humid" climate) in the introduction.*

**We agree with your suggestion. However, the mean annual potential evapotranspiration (ETp) is unknown for the study sites. The MAT and MAP are listed in a recent publication (Scheibe et al., 2023). Therefore, we added the following to the text (new text in green; Line 72-73):**

**One profile is located in a Mediterranean (mean annual temperature: 14.9 °C, mean annual precipitation: 436 mm $yr^{-1}$) and another in a humid climate zone (mean annual temperature: 14.1 °C, mean annual precipitation: 1084 mm $yr^{-1}$) (Scheibe et al., 2023), and both developed from weathering of granitic rock.**

*L94/95 + 110/114: To allow for comparison between uplift and denudation rates, can you express all values in the same units? (either mm yr-1 or t km-2 yr-1)?*

**By assuming a density of 2.6 g cm$^{-3}$ we converted the unit [t km$^{-2}$ yr$^{-1}$] to [mm yr$^{-1}$] and added this information in Line 95 and 113 (new text in green):**
**"The soil denudation rate in the nearby La Campana National Park is 53.7 ± 3.4 (S-facing slope) to 69.2 ± 4.6 t km$^{-2}$ yr$^{-1}$ (N-facing slope) (Oeser et al., 2018) or assuming a material density of 2.6 g cm$^{-3}$, 0.024 mm yr$^{-1}$ on average."**
**"The soil denudation rate in the nearby Nahuelbuta National Park ranges between 17.7 ± 1.1 (N-facing slope) to 47.5 ± 3.0 t km$^{-2}$ yr$^{-1}$ (S-facing slope) (Oeser et al., 2018) or assuming a material density of 2.6 g cm$^{-3}$, 0.013 mm yr$^{-1}$ on average."**

*L149 & L153-157: The equation for the CDF is somewhat different than Eq. 1 in Riebe et al. (2003)? Can you briefly explain the difference in the text?*

**Thank you very much for spotting this mistake. We calculated the CDF values correctly, but the terms in the brackets of Equation 1 (Line 149) are incorrect as the subscript "N" of Zr already indicates the normalisation. The modification of the CDF equation is explained in the data publication. To make that clear, we added "(see Hampl et al., 2022b)" in the explanation of the equation (Line 156, 157) and deleted the brackets in Equation 1 as well as the explanation of sum$^w$, LOI$^w$, sum$^b$ and LOI$^b$ (new text in green):**
**"Zr$_N^b$ = zirconium content of the bedrock normalized to a LOI-free sum of 100 % (see Hampl et al., 2022b), Zr$^w$ = concentration of Zr in the weathered sample, Zr$_N^w$ = zirconium content of the weathered sample normalized to a LOI-free sum of 100 % (see Hampl et al., 2022b)."**

*Oxalate- and dithionite-extractable Fe, Si, and Al are often noted with subscripts "o" or "d" in the literature.*

**Thank you for this hint. As our abbreviations do not cause any confusion, we would like to keep them and do not want to change "did" to "d" and "ox" to "o".**

*L335-337: Are these values wt.% of Mg-oxides?*

**Thank you for the question. As shown in Figure 7 the unit of the magnetite content is vol.%. We will add "vol." to the values in question (new text in green, Line 335-337):**

**"A relative magnetite enrichment was detected in the uppermost 40 cm of the LC profile (1–1.6 vol.%) whereas the rest of the profile shows approximately constant magnetite contents (mean ~0.9 vol.%) close to the value of the investigated bedrock (0.94 vol.%; Fig. 7c)."**

*L513: Clay migration (illuviation) can also play a role in the decrease of porosity.*

**Thank you for this comment. Since we do not see evidence for illuviation, we prefer not to add this topic.**

*Fig1: would be useful to have latlong added to the map -> panel(a)?*

**The latitude and longitude are added to panel (a) of Figure 1.**

*Fig2 & 3 & 4: Very nice figures*

**Thank you.**

*Fig7: Does the "pebble-size" material correspond to the fraction of the sample that is coarser than 2 mm? If so, you could also refer to "gravel" following the Udden-Wentworth grain-size scale.*

**The term is changed in Figure 7 and the sentence mentioning the result is adapted to (new text in green, Line 339):**
**"The soil pit profile of NA is characterized by a much higher gravel- and silt/clay-size content compared to LC (Fig. 7d)."**
**We also changed it in the corresponding data publication.**

*Fig10: Schematic summary is a great addition to the paper.*

**Thank you.**

**References**

Hampl, F. J., Schiperski, F., Schwerdhelm, C., Stroncik, N., Bryce, C., von Blanckenburg, F., and Neumann, T.: Mineralogical and geochemical data of two weathering profiles in a Mediterranean and a humid climate region of the Chilean Coastal Cordillera, GFZ Data Services [data set], https://doi.org/10.5880/fidgeo.2022.035, 2022b.

Oeser, R. A., Stroncik, N, Moskwa, L.-M., Bernhard, N., Schaller, M., Canessa, R., van den Brink, L., Köster, M., Brucker, E., Stock, S., Fuentes, J. P., Godoy, R., Matus, F. J., Oses Pedraza, R., Osses McIntyre, P., Paulino, L., Seguel, O., Bader, M. Y., Boy, J., Dippold, M. A., Ehlers, T. A., Kühn, P., Kuzyakov, Y., Leinweber, P., Scholten, T., Spielvogel, S., Spohn, M., Übernickel, K., Tielbörger, K., Wagner, D., and von Blanckenburg, F.: Chemistry and microbiology of the Critical Zone along a steep climate and vegetation gradient in the Chilean Coastal Cordillera, CATENA, 170, 183–203, https://doi.org/10.1016/j.catena.2018.06.002, 2018.

Reis, F. D. A. A., and S. L. Brantley (2017), Models of transport and reaction describing weathering of fractured rock with mobile and immobile water,J. Geophys.Res. Earth Surf.,122, 735–757, doi: 10.1002/2016JF004118.

Scheibe, A., Sierra, C. A., Spohn, M.: Recently fixed carbon fuels microbial activity several meters below the soil surface. Biogeosciences, 20, 827-838, https://doi.org/10.5194/bg-20-827-2023, 2023.

Schoonejans, J., Vanacker, V., Opfergelt, S., Ameijeiras-Mariño, Y., Christl, M.: Kinetically limited weathering at low denudation rates in semiarid climatic conditions. J. Geophys. Res. Earth Surf., 121, 336–350, https://doi.org/10.1002/2015JF003626, 2016.

Übernickel, K., Ehlers, T. A., Ershadi, M. R., Paulino, L., Fuentes Espoz, J.-P., Maldonado, A., Oses-Pedraza, R., and von Blanckenburg, F.: Time series of meteorological station data in the EarthShape study areas in the Coastal Cordillera, Chile, GFZ Data Services [data set], https://doi.org/10.5880/fidgeo.2020.043, 2020.

**Additions and changes to the manuscript**

This is a structured compilation of the additions (in green) and changes to the manuscript. The line numbers we give here refer to the revised version (with accepted changes) of our manuscript.

Line 72-75: One profile is located in a Mediterranean (mean annual temperature: 14.9 °C, mean annual precipitation: 436 mm yr$^{-1}$) and another in a humid climate zone (mean annual temperature: 14.1 °C, mean annual precipitation: 1084 mm yr$^{-1}$) (Scheibe et al., 2023), and both developed from weathering of granitic rock.

Line 92-94: Records of long-term meteorological data (e.g., precipitation at ground level, soil water content, air temperature, relative humidity) from a weather station near the study site can be found in Übernickel et al. (2020).

Line 98-100: The soil denudation rate in the nearby La Campana National Park is 53.7 ± 3.4 (S-facing slope) to 69.2 ± 4.6 t km$^{-2}$ yr$^{-1}$ (N-facing slope) (Oeser et al., 2018) or assuming a material density of 2.6 g cm$^{-3}$, 0.024 mm yr$^{-1}$ on average.

Line 111-113: Records of long-term meteorological data (e.g., precipitation at ground level, soil water content, air temperature, relative humidity) from a weather station near the study site can be found in Übernickel et al. (2020).

Line 119-120: The soil denudation rate in the nearby Nahuelbuta National Park ranges between 17.7 ± 1.1 (N-facing slope) to 47.5 ± 3.0 t km$^{-2}$ yr$^{-1}$ (S-facing slope) (Oeser et al., 2018) or assuming a material density of 2.6 g cm$^{-3}$, 0.013 mm yr$^{-1}$ on average."

Line 156:

$$\mathrm{CDF} = 1 - \frac{Zr_N^b \cdot (sum^w - LOI^w)}{Zr_N^w \cdot (sum^b - LOI^b)}$$

Line 160-163:  X$^b$ = concentration of element X in the bedrock, X$^w$ = concentration of element X in the weathered sample, Zr$^b$ = concentration of Zr in the bedrock, Zr$_N^b$ = zirconium content of the bedrock normalized to a LOI-free sum of 100 % (see Hampl et al.,

2022b), $Zr^w$ = concentration of Zr in the weathered sample, $Zr_N^w$ = zirconium content of the weathered sample normalized to a LOI-free sum of 100 % (see Hampl et al., 2022b).

Line 341-343: A relative magnetite enrichment was detected in the uppermost 40 cm of the LC profile (1–1.6 vol.%) whereas the rest of the profile shows approximately constant magnetite contents (mean ~0.9 vol.%) close to the value of the investigated bedrock (0.94 vol.%; Fig. 7c).

Line 345: The soil pit profile of NA is characterized by a much higher gravel- and silt/clay-size content compared to LC (Fig. 7d).

Line 425-426: This would also concur with the finding that water availability in the soil and soil residence time are the limiting factors for weathering processes in dry environments (Schoonejans et al., 2016).

Line 608-609: The authors would also like to thank Prof. Dr. Peter Finke, Prof. Dr. Veerle Vanacker and Prof. Dr. Susan L. Brantley for their valuable comments and suggestions that greatly improved the manuscript.